# THE GEOMETRY OF DEEP GENERATIVE IMAGE MODELS AND ITS APPLICATIONS

**Binxu Wang**
Department of Neuroscience
Washington University in St Louis
St Louis, MO, USA
`binxu.wang@wustl.edu`

**Carlos R. Ponce**
Department of Neuroscience
Washington University in St Louis
St Louis, MO, USA
`crponce@wustl.edu`

## ABSTRACT

Generative adversarial networks (GANs) have emerged as a powerful unsupervised method to model the statistical patterns of real-world data sets, such as natural images. These networks are trained to map random inputs in their latent space to new samples representative of the learned data. However, the structure of the latent space is hard to intuit due to its high dimensionality and the non-linearity of the generator, limiting the usefulness of the models. Understanding the latent space requires a way to identify input codes for existing real-world images (inversion), and a way to identify directions with known image transformations (interpretability). Here, we use a geometric framework to address both issues simultaneously. We develop an architecture-agnostic method to compute the Riemannian metric of the image manifold created by GANs. The eigen-decomposition of the metric isolates axes that account for different levels of image variability. An empirical analysis of several pretrained GANs shows that image variation around each position is concentrated along surprisingly few major axes (the space is highly anisotropic) and the directions that create this large variation are similar at different positions in the space (the space is homogeneous). We show that many of the top eigenvectors correspond to interpretable transforms in the image space, with a substantial part of eigenspace corresponding to minor transforms which could be compressed out. This geometric understanding unifies key previous results related to GAN interpretability. We show that the use of this metric allows for more efficient optimization in the latent space (e.g. GAN inversion) and facilitates unsupervised discovery of interpretable axes. Our results illustrate that defining the geometry of the GAN image manifold can serve as a general framework for understanding GANs.

## 1 BACKGROUND

Generative adversarial networks (GANs) learn patterns that characterize complex datasets, and subsequently generate new samples representative of that set. In recent years, there has been tremendous success in training GANs to generate high-resolution and photorealistic images (Karras et al., 2017; Brock et al., 2018; Donahue & Simonyan, 2019; Karras et al., 2020). Well-trained GANs show smooth transitions between image outputs when interpolating in their latent input space, which makes them useful in applications such as high-level image editing (changing attributes of faces), object segmentation, and image generation for art and neuroscience (Zhu et al., 2016; Shen et al., 2020; Pividori et al., 2019; Ponce et al., 2019). However, there is no systematic approach for understanding the latent space of any given GAN or its relationship to the manifold of natural images.

Because a generator provides a smooth map onto image space, one relevant conceptual model for GAN latent space is a Riemannian manifold. To define the structure of this manifold, we have to ask questions such as: are images homogeneously distributed on a sphere? (White, 2016) What is the structure of its tangent space — do all directions induce the same amount of variance in image transformation? Here we develop a method to compute the metric of this manifold and investigate its geometry directly, and then use this knowledge to navigate the space and improve several applications.

To define a Riemannian geometry, we need to have a smooth map and a notion of distance on it, defined by the metric tensor. For image applications, the relevant notion of distance is in image space rather than code space. Thus, we can pull back the distance function from the image space onto the latent space. Differentiating this distance function on latent space, we will get a differential geometric structure (Riemannian metric) on the image manifold. Further, by computing the Riemannian metric at different points (i.e. around different latent codes), we can estimate the anisotropy and homogeneity of this manifold.

The paper is organized as follows: first, we review the previous work using tools from Riemannian geometry to analyze generative models in section 2. Using this geometric framework, we introduce an efficient way to compute the metric tensor $H$ on the image manifold in section 3, and empirically investigate the properties of $H$ in various GANs in section 4. We explain the properties of this metric in terms of network architecture and training in section 5. We show that this understanding provides a unifying principle behind previous methods for interpretable axes discovery in the latent space. Finally, we demonstrate other applications that this geometric information could facilitate, e.g. gradient-free searching in the GAN image manifold in section 6.

## 2 RELATED WORK

**Geometry of Deep Generative Model** Concepts in Riemannian geometry have been recently applied to illuminate the structure of latent space of generative models (i.e. GANs and variational autoencoders, VAEs). Shao et al. (2018) designed algorithms to compute the geodesic path, parallel transport of vectors and geodesic shooting in the latent space; they used finite difference together with a pretrained encoder to circumvent the Jacobian computation of the generator. While promising, this method did not provide information of the metric directly and could not be applied to GANs without encoders. Arvanitidis et al. (2017) focused on the geometry of VAEs, deriving a formula for the metric tensor in order to solve the geodesic in the latent space; this worked well with shallow convolutional VAEs and low-resolution images (28 x 28 pixels). Chen et al. (2018) computed the geodesic through minimization, applying their method on shallow VAEs trained on MNIST images and a low-dimensional robotics dataset. In the above, the featured methods could only be applied to neural networks without ReLU activation. Here, our geometric analysis is architecture-agnostic and it's applied to modern large-scale GANs (e.g. BigGAN, StyleGAN2). Further, we extend the pixel L2 distance assumed in previous works to any differentiable distance metric.

## 3 METHODS

**Formulation** A generative network, denoted by $G$, is a mapping from latent code $\boldsymbol{z}$ to image $I$, $G : \mathbb{R}^n \to \mathcal{I} = \mathbb{R}^{H \times W \times 3}, \boldsymbol{z} \mapsto I$. Borrowing the language of Riemannian geometry, $G(\boldsymbol{z})$ parameterizes a submanifold in the image space with $\boldsymbol{z} \in \mathbb{R}^n$. Note for applications in image domain, we care about distance in the image space. Thus, given a distance function in image space $D : \mathcal{I} \times \mathcal{I} \to \mathbb{R}_+, (I_1, I_2) \mapsto L$, we can define the distance between two codes as the distance between the images they generate, i.e. pullback the distance function to latent space through $G$. $d : \mathbb{R}^n \times \mathbb{R}^n \to \mathbb{R}_+, d(\boldsymbol{z}_1, \boldsymbol{z}_2) := D(G(\boldsymbol{z}_1), G(\boldsymbol{z}_2))$.

The Hessian matrix (second order partial derivative) of the squared distance function $d^2$ can be seen as the metric tensor of the image manifold (Palais, 1957). The intuition behind this is as follows: consider the squared distance to a fixed reference vector $\boldsymbol{z}_0$ as a function of $\boldsymbol{z}$, $f_{\boldsymbol{z}_0}(\boldsymbol{z}) = d^2(\boldsymbol{z}_0, \boldsymbol{z})$. Obviously, $\boldsymbol{z} = \boldsymbol{z}_0$ is a local minimum of $f_{\boldsymbol{z}_0}(\boldsymbol{z})$, thus $f_{\boldsymbol{z}_0}(\boldsymbol{z})$ can be locally approximated by a positive semi-definite quadratic form $H(\boldsymbol{z}_0)$ as in Eq.1. This matrix induces an inner product and defines a vector norm, $\|\boldsymbol{v}\|_H^2 = \boldsymbol{v}^T H(\boldsymbol{z}_0) \boldsymbol{v}$. This squared vector norm approximates the squared image distance, $d^2(\boldsymbol{z}_0, \boldsymbol{z}_0 + \delta\boldsymbol{z}) \approx \|\delta\boldsymbol{z}\|_H^2 = \delta_{\boldsymbol{z}}^T H(\boldsymbol{z}_0) \delta\boldsymbol{z}$. Thus, this matrix encodes the local distance information on the image manifold up to second order approximation. This is the intuition behind Riemannian metric. In this article, the terms "metric tensor" and "Hessian matrix" are used interchangeably. We will call $\alpha_H(\boldsymbol{v}) = \boldsymbol{v}^T H \boldsymbol{v} / \boldsymbol{v}^T \boldsymbol{v}$ the approximate speed of image change along $\boldsymbol{v}$ as measured by metric $H$.

$$d^2(\boldsymbol{z}_0, \boldsymbol{z}) \approx \delta\boldsymbol{z}^T \frac{\partial^2 d^2(\boldsymbol{z}_0, \boldsymbol{z})}{\partial \boldsymbol{z}^2}|_{\boldsymbol{z}_0} \delta\boldsymbol{z}, \quad H(\boldsymbol{z}_0) := \frac{\partial^2 d^2(\boldsymbol{z}_0, \boldsymbol{z})}{\partial \boldsymbol{z}^2}|_{\boldsymbol{z}_0} \tag{1}$$

**Numerical Method**   As defined above, the metric tensor $H$ can be computed by doubly differentiating the squared distance function $d^2$. Here we use a convolutional neural network (CNN)-based distance metric, LPIPS (Zhang et al., 2018), as it has been demonstrated to approximate human perceptual similarity judgements. The direct method to compute Hessian is by building a computational graph towards the gradient $\boldsymbol{g}(\boldsymbol{z}) = \partial_{\boldsymbol{z}} d^2|_{\boldsymbol{z}=\boldsymbol{z}_0}$ and then computing the gradient towards each element in $\boldsymbol{g}(\boldsymbol{z})$. This method computes $H$ column by column, therefore its time complexity is proportional to the latent-space dimension $n$ and the backpropagation time through this graph.

For situations when direct backpropagation is too slow (e.g. FC6GAN, StyleGAN2), we developed an approximation method to compute the major eigen-dimensions of the Hessian more efficiently. These top eigen-pairs are useful in applications like optimization and exploration; moreover, they form the best low-rank approximation to the Hessian. As we will later discover, the spectra of these Hessians have a fast decay, thus far less than $n$ eigenvectors are required to approximate them, cf. Sec 4. As a matrix, the Hessian is a linear operator, which could be defined as long as one can compute the Hessian vector product (HVP). Since the gradient to $\boldsymbol{z}$ commutes with inner product with $\boldsymbol{v}$, HVP can be rewritten as the gradient to $\boldsymbol{v}^T \boldsymbol{g}$, or the directional derivative to the gradient $\boldsymbol{v}^T \partial_{\boldsymbol{z}} \boldsymbol{g}$ (Eq.2). The first form $\partial_{\boldsymbol{z}}(\boldsymbol{v}^T \boldsymbol{g})$ is easy to compute in reverse-mode auto-differentiation, and the directional derivative is easy to compute in forward-mode auto-differentiation (or finite differencing). Then, Lanczos iteration is applied to the HVP operator defined in these two ways to solve the largest eigen pairs, which can reconstruct an approximate Hessian matrix. The iterative algorithm using the two HVP definitions are termed Backward Iteration and Forward Iteration respectively. For details and efficiency comparison, see Appendix A.2.

$$HVP : \boldsymbol{v} \mapsto H\boldsymbol{v} = \partial_{\boldsymbol{z}}(\boldsymbol{v}^T \boldsymbol{g}(\boldsymbol{z})) = \boldsymbol{v}^T \partial_{\boldsymbol{z}} \boldsymbol{g}(\boldsymbol{z}) \approx (\boldsymbol{g}(\boldsymbol{z} + \epsilon\boldsymbol{v}) - \boldsymbol{g}(\boldsymbol{z} - \epsilon\boldsymbol{v}))/2\|\epsilon\boldsymbol{v}\| \qquad (2)$$

Note a similar computational method has been employed to understand the optimization landscape of deep neural networks recently (Ghorbani et al., 2019), although it has not been applied towards the geometry of latent space of GANs before.

**Connection to Jacobian**   This formulation and computation of the Riemannian metric is generic to any mapping into a metric space. Consider a mapping $\phi(\boldsymbol{z}) : \mathbb{R}^n \to \mathbb{R}^M$, which could be the feature map of a layer in the GAN, or a CNN processing the generated image. We can pull back the squared L2 distance and metric from $\mathbb{R}^M$, $d_\phi^2(\boldsymbol{z}_1, \boldsymbol{z}_2) = \frac{1}{2}\|\phi(\boldsymbol{z}_1) - \phi(\boldsymbol{z}_2)\|_2^2$, and define a manifold. The metric tensor $H_\phi$ of this manifold can be derived as Hessian of $d_\phi^2$. Note, there is a simple relationship between the Hessian of $d_\phi^2$, $H_\phi$ and the Jacobian of $\phi$, $J_\phi$ (Eq. 3). Through this we know the eigenvalues and eigenvectors of the Hessian matrix $H_\phi$ correspond to the squared singular values and right singular vectors of the Jacobian $J_\phi$. This allows us to examine the geometry of any representation throughout the GAN, and analyze how the geometry in the image space builds up.

$$H_\phi(\boldsymbol{z}_0) = \frac{\partial^2}{\partial \boldsymbol{z}^2} \frac{1}{2}\|\phi(\boldsymbol{z}_0) - \phi(\boldsymbol{z})\|_2^2|_{\boldsymbol{z}_0} = J_\phi(\boldsymbol{z}_0)^T J_\phi(\boldsymbol{z}_0) \qquad (3)$$

$$\boldsymbol{v}^T H_\phi(\boldsymbol{z}_0)\boldsymbol{v} = \|J_\phi(\boldsymbol{z}_0)\boldsymbol{v}\|^2, \quad J_\phi(\boldsymbol{z}_0) = \partial_{\boldsymbol{z}} \phi(\boldsymbol{z})|_{\boldsymbol{z}_0} \qquad (4)$$

In this work, we use LPIPS, which defines image distance based on the squared L2 distance of the first few layers of a pretrained CNN. If we concatenate the activations and denote this representational map by $\varphi(I) : \mathcal{I} \to \mathbb{R}^F$, then the metric tensor of the image manifold can be derived from the Jacobian of the composite of the generator and the representation map $\varphi$, $H(\boldsymbol{z}) = J_{\varphi \circ G}^T J_{\varphi \circ G}, J_{\varphi \circ G} = \partial_{\boldsymbol{z}} \varphi(G(\boldsymbol{z}))$. This connection is crucial for understanding how geometry depends on the network architecture.

## 4   EMPIRICAL OBSERVATIONS

Using the above method, we analyzed the geometry of the latent space of the following GANs: DCGAN (Radford et al., 2015), DeePSiM/FC6GAN (Dosovitskiy & Brox, 2016), BigGAN (Brock et al., 2018), BigBiGAN (Donahue & Simonyan, 2019), Progressive Growing of GANs (PGGAN) (Karras et al., 2017), StyleGAN 1 and 2 (Karras et al., 2019; 2020) - model specifications reviewed in Sec. A.3. These GANs are progressively deeper and more complex, and some employ a style-based architecture instead of the conventional DCGAN architecture (e.g. StyleGAN1,2). This diverse set of models allowed us to test the broad applicability of this new approach. In the following sections, "top" and "bottom" eigenvectors refer to the eigenvectors with large and small eigenvalues.

**Top Eigenvectors Capture Significant Image Changes.** In differential geometry, a metric tensor $H$ captures an infinitesimal notion of distance. To determine whether this quantity represents evident image changes, we randomly picked a latent code $z_0$, then computed the metric tensor $H(z_0)$ and its eigendecomposition $H(z_0) = \sum_i \lambda_i v_i v_i^T$. Then we explored linearly in the latent space[1] along the eigenvectors $G(z_0 + \mu_i v_i)$. We found that images changed much faster when moving along top than along bottom eigenvectors, both per visual inspection and LPIPS (Fig.1). More intriguingly, eigenvectors at different ranks encoded qualitatively different types of changes; for example, in BigGAN noise space, the top eigenvectors encoded head direction, proximity and size; while lower eigenvectors encoded background changes, shading or much more subtle pixel-wise changes. Moreover, PGGAN and StyleGANs trained on the face dataset (celebA,FFHQ) have top eigenvectors that represent similar interpretable transforms of faces, such as head direction, sex or age (Fig.10). These observations raised the possibility that top eigenvectors also captured *perceptually relevant* changes: we tested this directly with positive results in Sec. 6.

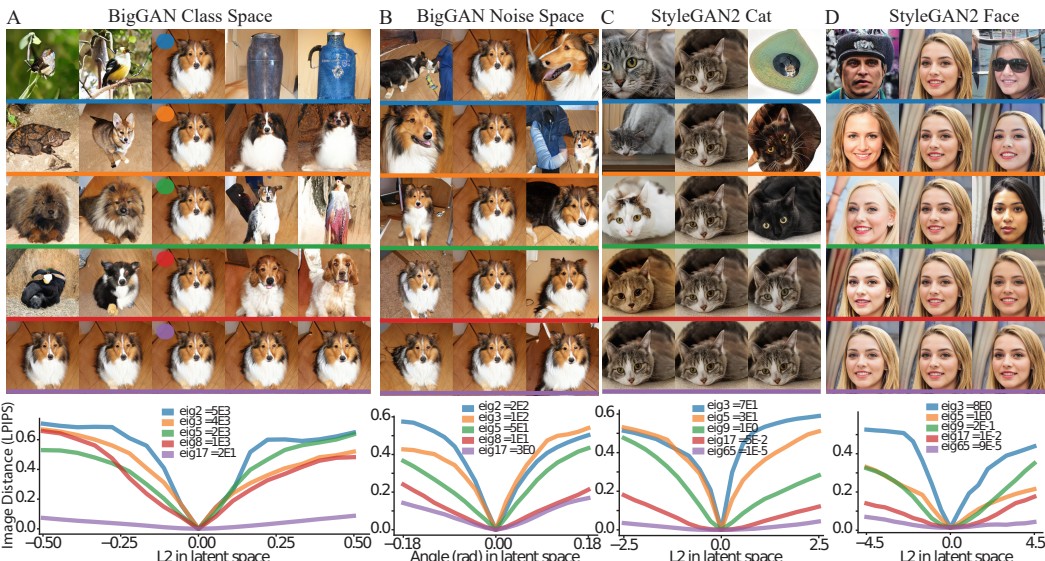

Figure 1: **Images change at different rates along top vs bottom eigenvectors**. Each panel (A-D) shows perturbations around a randomly chosen reference image (center column); each row shows perturbations introduced by moving along each of five eigenvectors; each contiguous column is separated by the same distance in latent space. Eigenvectors are shown in descending order of their eigenvalues. Line plots under each montage show the LPIPS image distance to the reference image as a function of positively and negatively perturbed distance along each eigenvector (x-axis). The rate of image change differed across eigenvectors; top vs bottom eigenvectors encoded changes such as object class, head direction, pose, color, shading or other subtle details (e.g. fur variations in panel A, bottom). Eigenvector rank and associated eigenvalue labeled in the legend.

**Spectrum Structure of GANs** To explore how eigenvalues were distributed, for each GAN, we randomly sampled 100-1000 $z$ in the latent space, used backpropagation to compute $H(z)$ and then performed the eigendecomposition. In Fig. 2, we plotted the mean and 90% confidence interval of the spectra and found that they spanned 5-10 orders of magnitude, with fast decays; each spectrum was dominated by a few eigenvectors with large eigenvalues. In other words, only a small fraction of dimensions were responsible for major image changes (Table 2), while most dimensions introduced nuanced changes (e.g. shading, background) — thus GAN latent spaces were highly anisotropic.

We found this anisotropy in every GAN we tested, which raises the question of why it has not been discussed more frequently. One possibility is that the statistical properties of high dimensionality create an illusion of isotropy. When traveling along a random direction $v$ in latent space, the approximate rate of image change $\alpha_H(v) = v^T H v / v^T v$ is a weighted average of all eigenvalues as in Eq. 9. In Sec A.6, we show analytically that the variance of $\alpha(v)$ across random directions will be $2/(n+2)$ times smaller than the variance among eigenvalues. For example, in BigGAN latent space (256 dimensions), the eigenvalues span over six orders of magnitude, while the $\alpha(v)$ for random

---

[1]For some spaces, we used spherical linear exploration (i.e. SLERP), where we restrict the vector to a sphere of certain norm. We project $v_i$ onto tangent space of $z_0$ and travel on the big circle from $z_0$ along $v_i$.

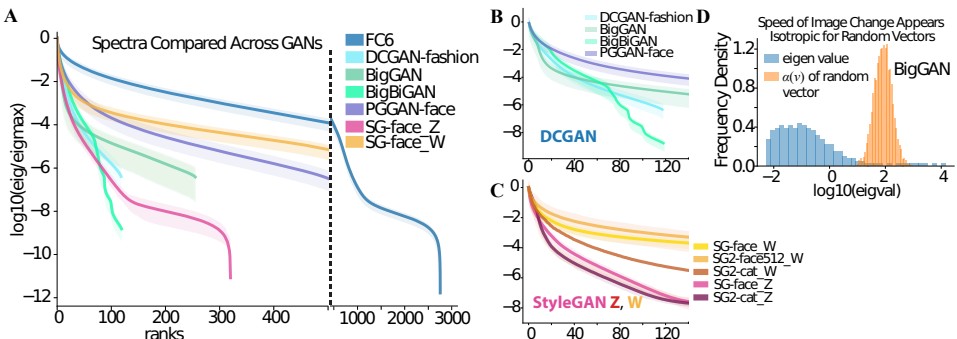

Figure 2: **Spectra of GANs**, shown as a function of individual model (A) and types of architecture (B, C). DCGAN-type (green-blue), Z and W space for StyleGAN (SG1, 2) (red, orange). Lines and shaded areas show the averaged spectra and the 5-95% percentile of each eigenvalue among samples (for quantification, see Table 2). D. Histogram of approximate speed of image change $\alpha(\boldsymbol{v})$ for eigenvectors and random directions, visualizing the "illusion of isotropy".

directions has a standard deviation less than one order of magnitude (Figs. 2, 6). Further, the center of this distribution was closer to the top of the spectrum, and thus provided a reasonable rate of change, while masking the existence of eigendimensions of extremely large and small eigenvalues.

**Global Metric Structure**    Because the metric $H(\boldsymbol{z})$ describes local geometry, the next question is how it varies at different positions in the latent space. We computed the metric $H(\boldsymbol{z})$ at randomly selected $\boldsymbol{z}$ and examined their similarity using a statistic adopted from Kornblith et al. (2019). In this statistic, we applied the eigenvectors $U_i = [u_1, ...u_n]$ from a metric tensor $H_i$ at position $\boldsymbol{z}_i$ to the metric tensor $H_j$ at $\boldsymbol{z}_j$, as $u_i^T H_j u_i$. These values formed a vector $\Lambda_{ij}$, representing the effects of metric $H_j$ on eigenvectors of $H_i$. Then we computed the Pearson correlation coefficient between $\Lambda_{ij}$ and the target eigenvalues, $\Lambda_j$, as $corr(\Lambda_j, \Lambda_{ij})$. This correlation measured the similarity of the action of metric tensors on eigenframes around different positions. As the spectrum usually spanned several orders of magnitude, we computed the correlation on the log scale $C_{ij}^{Hlog}$, where the eigenvalues distribute more uniformly.

$$\Lambda_{ij} = diag(U_i^T H(\boldsymbol{z}_j)U_i) \qquad (5)$$

$$C_{ij}^H = corr(\Lambda_{ij}, \Lambda_j), \; C_{ij}^{Hlog} = corr(\log(\Lambda_{ij}), \log(\Lambda_j)) \qquad (6)$$

Using this correlation statistic, we computed the consistency of the metric tensor across hundreds of positions within each GAN latent space. As shown in Fig. 3C, the average correlation between eigenvalues and vHv values of two points $C_{ij}^{Hlog}$ was 0.934 in BigGAN. For DCGAN-type architecture, mean correlations on the log scale ranged from 0.92-0.99; for StyleGAN-1,2, 0.64-0.73 in the Z space, and 0.87-0.89 in the W space (Fig. 3D, Tab.4). Overall, this shows that the local directions that induce image changes of different orders of magnitude are highly consistent at different points in the latent space. Because of this, the notion of a "global" Hessian makes sense, and we estimated it for each latent space by averaging the Hessian matrices at different locations.

**Implication of the Null Space**    As the spectra have a large portion of small eigenvalues and the metric tensors are correlated in space, the bottom eigenvectors should create a global subspace, in which latent traversal will result in small or even imperceptible changes in the image. This is supported by our perceptual study, as over half of the subjects cannot see any change in image when latent vector move in bottom eigenspace. (Sec. 6). This perceptually "null" space has implications about exploration in the GAN space and interpretable axes discovery. As $G(\boldsymbol{z} + \boldsymbol{v}) \approx G(\boldsymbol{z})$, if one axis $\boldsymbol{u}$ encodes an interpretable transform $G(\boldsymbol{z}) \to G(\boldsymbol{z} + \boldsymbol{u})$, then shifting this vector by a vector in the null space $\boldsymbol{v}$ will still result in an interpretable axis $G(\boldsymbol{z}) \to G(\boldsymbol{z} + \boldsymbol{v} + \boldsymbol{u}) \approx G(\boldsymbol{z} + \boldsymbol{u})$. Thus, each interpretable axis have a family of "equivalent" axes which encode similar transforms, differing from each other by a vector in "null" space. However, adding component $\boldsymbol{v}$ in the null space will decrease the rate of image change along that axis. In this sense, the vectors using a smallest step size to achieve that transform should be the "purest" axes of the family. Further, the cosine angle between two interpretable axes may not represent the similarity of the transforms they encode. A large angle can be found between two axes of the same family but at different image traversal speed. We compared the axes from previous works in A.9 and observed that projecting out a large part of their axes did not affect the semantics it encoded (Fig. 8).

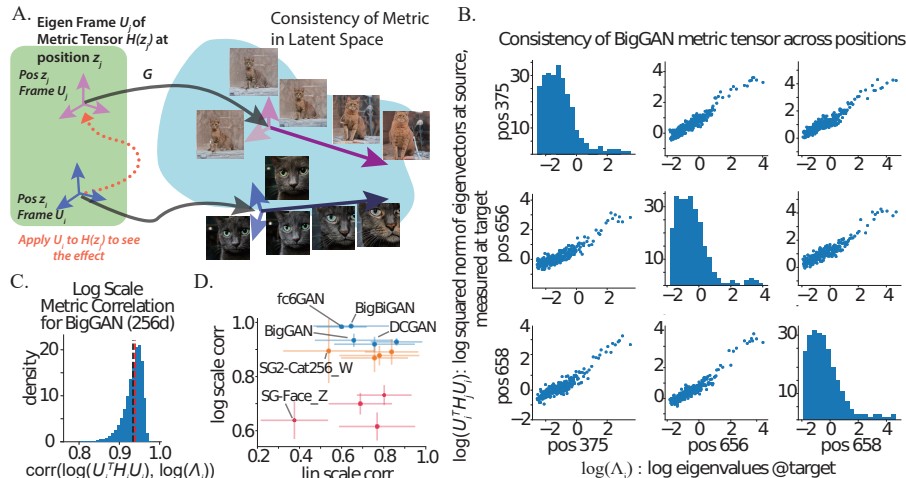

Figure 3: **Globalization of metric structure**: A. Schematic of the geometric picture. In the latent space (green area), the metric eigen frames at each point (blue and violet) are mapped to transformations in image space (blue area); the length and saturation of image-space vectors represent the eigenvalue (i.e. amplification factor) of $G$. We show that the top eigen space are relatively aligned at different positions. B. Distributions of $\log(\Lambda_{ij})$ and $\log(\Lambda_j)$, showing the action of metric are correlated at 3 different points. C. Histogram of correlation $C_{ij}^{Hlog}$ between all pairs among 1000 points in BigGAN space. D. Comparison of correlation values on linear and log scales for different GAN models. DCGAN-type (blue), Z and W space for StyleGAN1,2 (red and orange).

## 5 MECHANISM

Above, we showed an intriguingly consistent geometric structure across multiple GANs. Next, we sought to understand how this structure emerged through network architecture and training.

To link the metric tensor to the generator architecture, it is helpful to highlight the relationship between the metric tensor and Jacobian matrix $H(z) = J_{\varphi \circ G}^T J_{\varphi \circ G}$ (Eq. 3). As the latent space gets warped and mapped onto image space, directions in latent spaces are scaled differently by the Jacobian of the map; specifically, directions that undergo the most amplification will become the top eigenvectors (Fig. 4A). As the Jacobian of the generator is a composition of Jacobians of each layer $d\psi_i$, the scaling effect on the image manifold is a product of the scaling effects of each intermediate layer. We can analyze the scaling effect of different layers $\psi_i$ by applying a set of vectors onto the metric tensors of these layers $H_{\psi_i}$. In BigGAN, when we apply the eigenvectors of the first few layers onto the metric of other layers, the top eigenvectors are still strongly amplified by subsequent layers, thus forming the top eigendimensions of the manifold. Of note, this is not true for a weight-shuffled control BigGAN: in that case, the top eigendimension of the first few layers was not particularly amplified on the image manifold, and vice versa (Fig. 4 B). This shows that the amplification effect of layers becomes more aligned through training, with the top eigenspace shared across layers. Further, as the amplification effects are not lined up across layers of weight shuffled networks, these networks should exhibit a more isotropic geometry on their image manifold. Indeed, we find their spectra to be flatter and the largest eigenvalue smaller (Fig. 7).

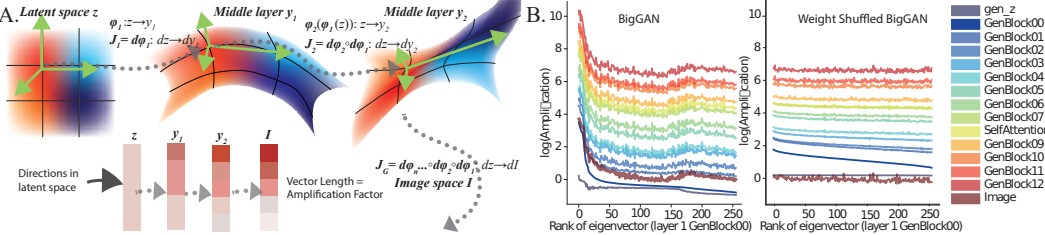

Figure 4: **Anisotropy is induced and maintained throughout the GAN architecture**. A. As latent space gets warped and mapped into image space, directions in latent spaces are scaled differently by the Jacobian of the map. B. Amplification of eigenvectors of the metric tensor of the first conv layer (GenBlock00) in all major layers in BigGAN. C. Same, but for weight-shuffled BigGAN.

## 6 APPLICATIONS

By defining the geometry of the latent space via the metric tensor, we gain an understanding of which directions in this space are more informative than others. This understanding leads to improvements in three applications: 1) finding human-interpretable axes in the latent space, 2) improving gradient-based optimizers, 3) accelerating gradient-free search.

**Interpretable Axes Discovery**   When users wish to manipulate generated images via their latent code, it is useful to reduce the number of variables needed to effectuate that manipulation. Our method provides a systematic way to compute the most informative axes (top eigenspace) in the latent space to use as variables, and the resulting eigenvalues can serve to compute appropriate step sizes along each corresponding axes. We visualized the image changes corresponding to the top eigenvectors in BigGAN, BigBiGAN, PGGAN, StyleGAN1,2 (Fig.1). We found many of these eigenvectors appeared to capture interpretable transformations like zooming, head direction and object position, consistent across reference image.

To test if this was apparent to people other than the authors, we conducted a study using Amazon's Mechanical Turk. We tested the perceptual properties of the axes identified by the metric tensor, including the top 10 eigenvectors, random vectors orthogonal to the top 15d eigenspace, and bottom 10 eigenvectors. Images were generated using four different GANs (PGGAN, BigGAN noise space, StyleGAN2-Cat and -Face), and were presented to 185 participants. In each trial, five randomly sampled reference images were perturbed along a given axis, and participants were asked if they could a) perceive a change, b) indicate an estimate of its magnitude [0%-100%] c) describe a common change in their own words and how many of the five images shared this change, c) indicate how similar were the 5 image changes (consistency, score of 1-9, 9 most similar) and finally, d) state how difficult it was to describe this change (difficulty score, scale of 1-9, 9 most difficult).

Only 48.5% of the subjects reported to see any change happen for bottom eigenvectors, while the fraction was 93.5% and 89.8% for top and orthogonal directions respectively. Further, when subjects observed some change, they reported that the image transformations induced by top eigenvectors were larger ($70.3\% \pm 0.6\%$) than those of orthogonal directions ($66.8\% \pm 0.9\%, P = 7.0 \times 10^{-4}$, 2 sample T-test) and than those of bottom eigenvectors ($61.5\% \pm 1.6\%, P = 2.1 \times 10^{-10}$). This was true even though we picked a step size in the top eigenspace that was 5-10 times smaller than in the orthogonal and bottom eigenspaces. Further, subjects reported the top 10 eigenvectors had a higher mean perceptual consistency score ($6.72 \pm 0.06$), $n = 929$ responses) than the orthogonal ($6.42 \pm 0.09, P = 5.8 \times 10^{-3}, n = 448$ responses) and bottom eigenvectors ($6.12 \pm 0.14, P = 1.3 \times 10^{-5}, n = 242$ responses). Participants reported that the top eigenvectors were easier to interpret ($4.91 \pm 0.08$) than the bottom eigenvectors ($5.69 \pm 0.14, P = 3.8 \times 10^{-6}$, albeit comparably to the orthogonal eigenvectors $4.82 \pm 0.11, P = 0.5$). Thus, overall we conclude that the Hessian eigenvectors not only capture informative axes of image transformations, but that these were also perceptually relevant, corresponding to similar semantic changes when applied to different reference vectors (Fig. 11) — axes interpretable not just in local sense, but in a global sense.

**Improving Gradient-Based GAN Inversion.**   For applications like GAN-assisted drawing and photo editing (Zhu et al., 2016; Shen et al., 2020), one crucial step is to find a latent code corresponding to a given natural image (termed GAN inversion). For this problem, one basic approach is to minimize the distance between a generated image and the target image $z^* = \arg \min_z D(G(z), I)$. Although second-order information (Hessian) is valuable in optimization, they are seldom used as they are expensive to compute and update. However, since we find that the local Hessians are highly correlated across the latent space, we can pre-compute it once for each latent space and use the global average Hessian to boost first-order optimization. As an example, ADAM is a first-order optimization algorithm that adapts the learning rate of each parameter separately according to the moments of gradients on that parameter (Kingma & Ba, 2014). It can be seen as a quasi-second order optimizer that approximates a diagonal Hessian matrix based on first-order information. However, if the true Hessian is far from diagonal, i.e. the space is anisotropic and the valley is not aligned with the coordinates, then this approximation could work poorly.

To test whether the metric can help overcome this problem, we used the eigenvectors of the global average Hessian to rotate the latent space; this orthogonal change of variables should make the Hessian more diagonal and thus accelerate ADAM. This method can be seen as a precondition-

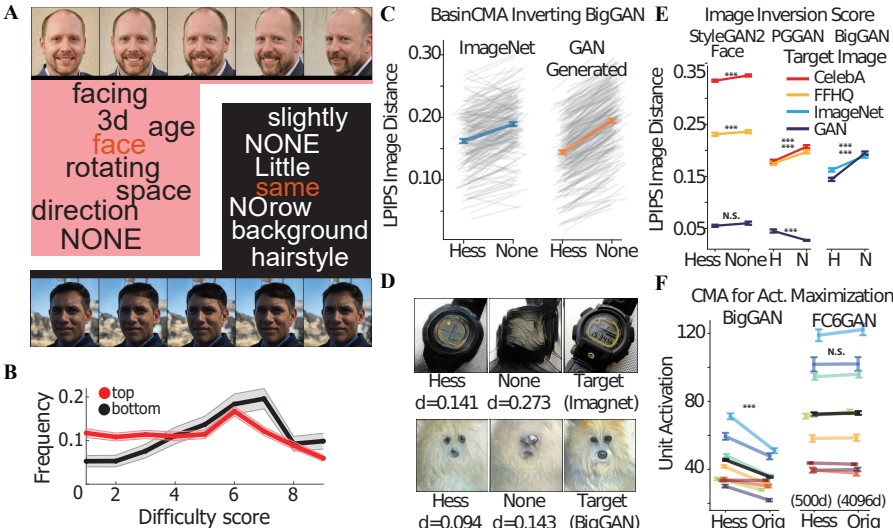

Figure 5: **Applications of the metric tensor A. Perceptual properties of eigenvectors**. Word cloud shows subjects' descriptions (N = 24) of the image transforms induced by the top- (red) vs. the bottom eigenvector (black) in StyleGAN2-Face. B. Distribution of difficulty scores associated with top- vs. bottom eigenvectors (red, black) across all four GANs for N=185 subjects. Lines show mean frequency $\pm$ standard error (per bootstrap). **C-E. Eigenbasis pre-conditioning improves GAN inversion.** C. BasinCMA with eigenbasis pre-conditioning (Hess,H) outperformed a method using normal basis (None,N) in inverting ImageNet and BigGAN generated images; D. Examples of fitted ImageNet and BigGAN images with our Hessian BasinCMA and original method (LPIPS distance below). E. Results for PGGAN and StyleGAN2 inverting samples from CelebA and FFHQ. **F. Hessian-CMAES (Hess) outperforms CMAES (Orig)** in maximizing CNN activation in BigGAN, and increases sampling efficiency in FC6GAN latent space. Each line represents a different layer of optimized units in AlexNet. *** denotes paired t-test $p < 1 \times 10^{-6}$

ing step which could be inserted into any pipeline involving ADAM. We tested this modification on the state-of-the-art algorithm for inverting BigGAN, i.e. BasinCMA (Huh et al., 2020), which interleaves ADAM and CMAES steps. We used our Hessian eigenbasis in the ADAM steps, and found that we could consistently lower the fitted distance to the target when inverting ImageNet and BigGAN-generated images (Fig. 5). Similarly, eigenbasis preconditioning consistently improved inversion of PGGAN and StyleGAN2-Face for real image sampled from both FFHQ and CelebA using ADAM method. In short, the understanding of homogeneity and anisotropy of the latent space can improve gradient-based optimization.

**Improving Gradient-Free Search in Image Space** In some domains, it is important to optimize objectives in the absence of a gradient, for example, in black-box attacks against image recognition systems via adversarial images, when searching for activity-maximizing stimuli for neurons in primate visual cortex, or when optimizing perceptual evaluation in the user (Ponce et al., 2019; Xiao & Kreiman, 2020; Chiu et al., 2020). These applications usually involve a low-dimensional parameter space (such as GANs) and an efficient gradient-free search algorithm, such as covariance matrix adaptation evolution strategy (CMAES). CMAES explores the latent space using a Gaussian distribution and adapts the shape of the Gaussian (covariance matrix) according to the search history and local landscape. However, online learning of a covariance matrix in high-dimensional space is computationally costly, and inaccurate knowledge of it can be detrimental to optimization. Here we applied the prior geometric knowledge of the space to build the covariance matrix instead of learning it from scratch. For example, as illustrated by natural gradient descent (Amari, 1998), one simple heuristic for optimizing on the image manifold is to move in smaller steps along dimensions that change the image faster to avoid overshoot. We built in this heuristic to improve CMAES, termed CMAES-Hessian. With our method, the search can be limited to the most informative directions, which should increase sampling efficiency; further, our method tunes the exploration step size in a way that is inversely proportional to the rate of image change. To test this approach, we applied our CMAES-Hessian algorithm to the problem of searching for activation maximizing stimuli for units in AlexNet (Nguyen et al., 2016) in the latent space of FC6GAN and BigGAN. We found that

the dimension of the search space could be reduced from 4096 to 500 for FC6GAN without impairing maximal activation values. Further, we found that CMAES-Hessian consistently led to higher activation values compared to the classic CMAES algorithm in BigGAN space (Fig. 5F).

# 7 DISCUSSION AND FUTURE DIRECTIONS

In this work, we developed an efficient and architecture-agnostic way to compute the geometry of the manifold learnt by generative networks. This method discovers axes accounting for the largest variation in image transformation, which frequently represent semantically interpretable changes. Subsequently, this geometric method can facilitate image manipulation, increase explainability, and accelerate optimization on the manifold (with or without gradients).

There have been multiple efforts directed at identifying interpretable axes in latent space using unsupervised methods, including (Ramesh et al., 2018; Härkönen et al., 2020; Shen & Zhou, 2020; Voynov & Babenko, 2020; Peebles et al., 2020). Our description of the connection between the metric tensor of the image manifold and the Jacobian matrices of intermediate layers unifies these previous results. As we have showed, the top right singular vectors of the weights (i.e. Jacobian) of the first few layers (as used in Shen & Zhou (2020)), correspond to the top eigenvectors of the metric tensor of the image manifold, and these usually relate to interpretable transforms. Similarly, the top principal components (PCs) of intermediate layer activations Härkönen et al. (2020) roughly correspond to the top left singular vectors of the Jacobian, thus also to the interpretable top eigenvectors of the metric on the image manifold. Likewise, Ramesh et al. (2018) also observed that the top right singular vectors of the Jacobian of $G$ are locally disentangled. Regarding Voynov & Babenko (2020) and Peebles et al. (2020), we empirically compared their interpretable axes and our eigenvectors, and found that in some of the GANs, the discovered axes have a significantly larger alignment with our top eigenspace and they are highly concentrated on individual top axes than expected from random mixing. We refer readers to Sec.A.9,A.10 and Fig.8,9 for further comparison.

Although we have answered how the anisotropy comes into being mechanistically, there remains the question of why it should exist at all. Anisotropy may result from gradient training: theoretical findings on deep-linear networks for classification show that gradient descent aligns the weights of layers, resulting in a highly anisotropic Jacobian (Ji & Telgarsky, 2018). Whether that analysis transfers to the setting of generative networks remains to be investigated.

Alternatively, assuming that a well-trained GAN faithfully represents the data distribution, this anisotropy may reveal the intrinsic dimensionality of the data manifold. Statistical dependencies of variation in real-world images imply that the images reside in a statistical manifold of much lower dimension. Further, among transformations that happen on this manifold, there will be some dimensions that transform images a lot and some that do not. In that sense, our method may be equivalent to performing a type of nonlinear PCA of the image space through the generator map. In fact, we have found that GANs trained on similar datasets (e.g. PGGAN, StyleGAN1,2 trained on the human face dataset CelebA,FFHQ) have top eigenvectors that represent the same set of transforms (e.g. head direction, gender, age; Fig. 10). This supports the "PCA" hypothesis, as these transformations may account for much of the pixelwise variability in face space; the GANs are able to learn to represent these transformations as linear directions, which our method can then identify.

This further raises the intriguing possibility that if the dataset is actually distributed on a lower dimensional space, one could learn generators with smaller latent spaces; or alternatively, it may be easier to learn generators with large latent spaces and reduce them after intensive training. These are questions worth exploring.

## ACKNOWLEDGMENTS

We appreciate the conceptual and technical inspirations from Dr. Timothy Holy (WUSTL). We are grateful for the constructive discussion with Zhengdao Chen (NYU), whose pointers to the relevant literature helped launch this work. We thank Hao Sun (CUHK) in providing experience for the submission and rebuttal process. We thank friends and colleagues Yunyi Shen (UW–Madison), Lingwei Kong (ASU) and Yuxiu Shao (ENS) who read and commented on our early manuscript. This work was supported by Washington University in St. Louis, the David and Lucille Packard Foundation, the McDonnell Center for Systems Neuroscience (predoctoral fellowship to B.W.), and a seed grant from the Center for Brains, Minds and Machines (CBMM; funded by NSF STC award CCF-1231216).

## AUTHOR CONTRIBUTIONS

B.W. conceptualized the project, designed the algorithm, developed the code base, performed the numerical experiment and analyzed the data. B.W. and C.R.P. interpreted the results. B.W. and C.R.P. designed the human MTurk task and analyzed the data. B.W. and C.R.P. prepared and revised the manuscript.

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

# A APPENDIX

## A.1 CONNECTION TO INFORMATION GEOMETRY

It is useful to compare our work to the "information geometry"(Amari, 2016) on the space of distributions. In that formulation, KL divergence is a pseudo-metric function on the space of distributions, and its Hessian matrix towards parameters of distribution is the Fisher information matrix. In information geometry, this Fisher information matrix could be considered as the metric on the manifold of distributions; this metric information can be further used to assist optimization on the manifold of distributions, termed natural gradient descent (Amari, 1998). In our formulation, the squared image difference function $D^2$ is analogous to this KL-divergence; the image $G(\boldsymbol{z})$ as parameterized by latent code $\boldsymbol{z}$ is analogous to the distribution $p_\theta$ parameterized by $\theta$. The metric tensor we computed is comparable to the Fisher information matrix in their setting. Thus our way of using metric information to assist optimization on manifold is analogous to natural gradient descent.

## A.2 METHODS FOR COMPUTING THE HESSIAN

One direct way to compute the Hessian of a given scalar function (e.g. squared distance $d^2$ in our case), is to compute $g_{\boldsymbol{z}_0}(\boldsymbol{z}) = \partial_{\boldsymbol{z}} d^2|_{\boldsymbol{z}=\boldsymbol{z}_0}$, create a computational graph from code $\boldsymbol{z}$ to the gradient $g_{\boldsymbol{z}_0}(\boldsymbol{z})$, and back propagate from gradient vector $g_{\boldsymbol{z}_0}(\boldsymbol{z})$ element by element. In this way the computational time is linear to time of a single backward pass times the latent space dimension $n$.

Given a large latent space or a deep network (e.g. 4096 dimensions in FC6GAN, or 512 in Style-GAN2), this method can be very slow. An efficient way is to use the Hessian vector product (HVP) operator and iterative eigenvalue algorithms like power iteration or Lancsoz iterations to solve the eigenvectors corresponding to eigenvalues of largest amplitudes. These largest eigen pairs create the best low rank approximation to the real Hessian matrix. Note that, to find the smallest amplitude eigen pairs, inverse Hessian vector product operator is required, which is much more expensive to compute. However, as the eigenspace with the smallest eigenvalues represent directions that do not change images much, the exact eigenvector does not matter. We can just define an arbitrary basis in the "null" space orthogonal to the eigenspace with large amplitude eigenvalues.

There are two ways to construct a HVP operator: one way uses the 2nd order computational graph from $\boldsymbol{z}$ to the gradient $g(\boldsymbol{z})$ to compute HVP by back-prop, i.e. $HVP_{backward}$; the other way uses finite difference on the first-order gradient to compute HVP i.e. $HVP_{forward}$. As it does not require backpropagation, a single operation of $HVP_{forward}$ is faster than $HVP_{backward}$ but it is less accurate and takes more iterations to converge. We use the ARPACK (Lehoucq et al., 1998) implementation of the Lanczos algorithm as our iterative eigenvalue solver.

$$HVP_{backward} : \boldsymbol{v} \mapsto \partial_{\boldsymbol{z}}(\boldsymbol{v}^T g(\boldsymbol{z})) \tag{7}$$

$$HVP_{forward} : \boldsymbol{v} \mapsto \frac{g(\boldsymbol{z} + \epsilon\boldsymbol{v}) - g(\boldsymbol{z} - \epsilon\boldsymbol{v})}{2\epsilon\|\boldsymbol{v}\|} \tag{8}$$

We termed the direct method Full BackProp (BP), the iterative method using $HVP_{backward}$ and $HVP_{forward}$ Backward Iteration and Forward Iteration respectively. We computed the Hessian at the same $\boldsymbol{z}_0$ using these three methods in different GANs and compared their temporal cost empirically in Table A.2.

Note, our method can be employed to compute the singular values and right singular vectors of the Jacobian from latent space towards any intermediate layer representation. To obtain the left singular vector, i.e. the change in representation or image space caused by the direction, we need to push forward the right singular vectors through the Jacobian, which is feasible through forward-mode autodiff or finite difference.

## A.3 SPECIFICATION OF GAN LATENT SPACE

The pretrained GANs used in the paper are from the following sources:

Table 1: **Computational Cost for Three Methods**: Computation time is measured on a GTX 1060 GPU. The iterative method has a variable runtime which depends on the number of eigenpairs required. In this table, we all use one half of the full dimension as the eigenpaired required, which is the largest we can require using ARPACK implementation of Lanczos. Thus these numbers should be seen as an upper limit of time for Backward and Forward Iteration method. A shallower or narrower network will result in faster computation time. For StyleGAN2 which has configurable depth and width, we use the config-f.

| | Dimension | Full BackProp Time | Backward Iter Time | Forward Iter Time |
|---|---|---|---|---|
| DCGAN | 120 | 12.5 | 13.4 | 6.9 |
| FC6 GAN | 4096 | 282.4 | 101.2 | 90.2 |
| BigGAN | 256 | 69.4 | 70.6 | 67 |
| BigBiGAN | 120 | 15.3 | 15.2 | 13.3 |
| PGGAN | 512 | 95.4 | 95.7 | 61.2 |
| StyleGAN | 512 | 112.8 | 110.5 | 64.5 |
| StyleGAN2* | 512 | 221 | 217 | 149 |

**DCGAN** model was obtained from torch hub `https://pytorch.org/hub/facebookresearch_pytorch-gan-zoo_dcgan/`. It's trained on 64 by 64 pixel fashion dataset. It has a 120d latent space, using Gaussian as latent space distribution.

**Progressive Growing GAN (PGGAN)** was obtained from torch hub `https://pytorch.org/hub/facebookresearch_pytorch-gan-zoo_pgan/` and we use the 256 pixel version. It's trained on celebrity faces dataset (CelebA). It has a 512d latent space, using Gaussian as latent space distribution.

**DeePSim, FC6GAN** model was re-written and translated into Pytorch, with weights obtained from official page `https://lmb.informatik.uni-freiburg.de/people/dosovits/code.html` of Dosovitskiy & Brox (2016). The architecture is designed to mirror that of AlexNet, and the FC6GAN model is trained to invert AlexNet's mapping from image to FC6 layer. Thus it has 4096d latent space. This model is highly expressive in fitting arbitrary pattern, but not particularly photorealistic.

**BigGAN** model was obtained through Hugging Face's translation of DeepMind's Tensorflow implementation `https://github.com/huggingface/pytorch-pretrained-BigGAN`, we use biggan-deep-256 version. It's trained on ImageNet dataset in a class conditional way. It has a 128d latent space called noise space, and a 128d embedding space for the 1000 classes called class space. The 2 vectors are concatenated and sent into the network. The distribution used to sample in noise space is truncated normal. Here we analyze the metric tensor computed in the concatenated 256d space (BigGAN) or in the 128d noise space or class space separately (BigGAN-noise, class).

**BigBiGAN** model was obtained via a translation of DeepMind's Tensorflow implementation `https://tfhub.dev/deepmind/bigbigan-resnet50/1`, we use bigbigan-resnet50 version. It's trained on ImageNet dataset in unconditioned fashion. It has a 120d latent space, using Gaussian as latent distribution. Note, the latent vector is split into six 20d trunks and sent into different parts of the model, which explains why the spectrum of BigBiGAN has the staircase form (in Fig. 2).

**StyleGAN** model was obtained via a translation of NVIDIA's Tensorflow implementation `https://github.com/rosinality/style-based-gan-pytorch`. We used the 256 pixel output. It has a 512d latent space called Z space, where the latent distribution is Gaussian distribution. This $Z$ distribution gets warped into another 512d latent space called W space, by a multi-layer perceptron. The latent vector $W$ is sent into a style-based generative network, in which the latent vector just modulates the feature maps in the conv layers, instead of serving as a spatial input as in DCGAN, FC6GAN, PGGAN.

**StyleGAN2** models are obtained via a translation of NVIDIA's Tensorflow implementation `https://github.com/rosinality/stylegan2-pytorch`. This is an improved version of StyleGAN: it also has a network mapping the 512d $Z$ to the $W$ space, and the style-based generative network. The various pre-trained models are fetched from `https://pythonawesome.`

Table 2: **Quantification of Spectra anisotropy** (Models marked with † are audio wave form generating GANs using different distance metric function.)

| | dimen | dim.99 | dim.999 | dim.9999 | dim.99999 |
|---|---|---|---|---|---|
| FC6 | 4096 | 297 | 502 | 661 | 848 |
| DCGAN-fashion | 120 | 17 | 35 | 65 | 97 |
| BigGAN | 256 | 10 | 53 | 149 | 224 |
| BigGAN_noise | 128 | 29 | 88 | 120 | 127 |
| BigGAN_class | 128 | 8 | 38 | 98 | 123 |
| BigBiGAN | 120 | 21 | 41 | 62 | 73 |
| PGGAN-face | 512 | 57 | 167 | 325 | 450 |
| StyleGAN-face_Z | 512 | 12 | 27 | 52 | 84 |
| StyleGAN2-face512_Z | 512 | 7 | 17 | 41 | 78 |
| StyleGAN2-face256_Z | 512 | 13 | 28 | 63 | 103 |
| StyleGAN2-cat_Z | 512 | 8 | 14 | 31 | 62 |
| StyleGAN-face_W | 512 | 124 | 355 | 480 | 507 |
| StyleGAN2-face512_W | 512 | 153 | 345 | 471 | 506 |
| StyleGAN2-face256_W | 512 | 157 | 350 | 473 | 506 |
| StyleGAN2-cat_W | 512 | 23 | 57 | 126 | 269 |
| WaveGAN_MSE† | 100 | 17 | 38 | 74 | 94 |
| WaveGAN_STFT† | 100 | 2 | 9 | 19 | 42 |

com/a-collection-of-pre-trained-stylegan-2-models-to-download.
More specifically StyleGAN2-Face256 and 512 are both trained on FFHQ dataset, while Face256 generate lower resolution images and use narrower conv layers. StyleGAN2-Cat is trained on LSUN cat dataset (Yu et al., 2015) at 512 resolution.

**WaveGAN** model is obtained from the repository `https://github.com/mostafaelaraby/wavegan-pytorch/`. Its architecture resembles that of DCGAN, but applied to the one dimensional wave form generation problem. We customly trained it on the wave forms of piano performance clips. It has a 100d latent space, using Gaussian as latent space distribution.

## A.4 QUANTIFICATION OF POWER DISTRIBUTION IN SPECTRA

We quantified the anisotropy of the space, i.e. the low rankness of the metric tensor in Table 2. To do this, we computed the number of eigenvalues needed to account for the 0.99, 0.999, 0.9999, 0.99999 fraction of the sum of all eigenvalues. This can be thought of as the minimal number of dimensions needed to achieve a low rank approximation of the Jacobian with 0.01, 0.001, 0.0001, 0.00001 residue in terms of the Frobenius norm.

There are a few interesting patterns we noticed in this table. For BigGAN, we noted that the class subspace is more low-ranked than the noise subspace, i.e. fewer directions could account for most of the changes across object classes than within classes. For StyleGAN 1 and 2, we analyzed the geometry of $Z$ space and $W$ space separately, and found that in all the models the metric in $W$ space is significantly more isotropic i.e. less low rank than $Z$ space. Thus, in this regard, the $z \to w$ mapping warped the spherical distribution in $Z$ space to an elongated one in $W$ space, but the mapping from $W$ space to image is still more isotropic.

## A.5 GEOMETRIC STRUCTURE IS ROBUST TO THE IMAGE DISTANCE METRIC

Our work used the LPIPS distance metric to compute the Riemannian metric tensor. To determine how much of the results depended on this choice of metric, we computed the metric tensor at the same hidden vector using different image distance functions, specifically a) structural similarity index measure (SSIM) and b) Mean Squared Error (MSE) in pixel space, which do not depend on CNN. We computed the Hessian at 100 random sampled vectors in BigGAN, Progressive Growing GAN (Face), StyleGAN2 (Face 256), using MSE, SSIM and LPIPS, and then compared their Hessian spectra and eigenvectors. We found that the entry-wise correlation across the Hessian matrices ($d^2$ elements) ranged from [0.94-0.99]. The correlation of eigenvalue spectra ranged from [0.987-

Table 3: **Comparison of Hessian Computed with Different Sample Dissimilarity Metric** $d$ We experimented with BigGAN, PGGAN and StyleGAN2 (FFHQ 256 resolution), we compared the Hessian computed by LPIPS and that by MSE or SSIM at 100 latent vectors. The statistics we showed are: element-wise Hessian correlation (H corr), eigen spectra correlation (eigval corr), the Hessian consistency measure $C^{Hlin}$ and $C^{Hlog}$. The linear regression between the log spectra of LPIPS the and that of the alternative (SSIM or MSE) yields the slope (slope) and intercept (intercept). The mean and standard deviation (in parenthesis) of the the 100 statistics are shown. In the last row, WaveGAN† is an audio generating GAN. We measured the similarity of the Hessian using MSE of wave forms (MSE) and MSE of spectrogram (STFT) as dissimilarity metric. Hessian computed using these two measures are less similar to each other.

| | | H corr | eigval corr | $C^{Hlin}$ | $C^{Hlog}$ | slope | intercept |
|---|---|---|---|---|---|---|---|
| BigGAN | MSE | 0.973(0.033) | 0.995(0.008) | 0.996(0.014) | 0.999(0.001) | 1.01(0.03) | 1.47(0.22) |
| | SSIM | 0.988(0.012) | 0.997(0.003) | 0.999(0.002) | 0.999(0.001) | 1.06(0.02) | -0.40(0.07) |
| PGGAN | MSE | 0.938(0.047) | 0.987(0.016) | 0.987(0.038) | 0.999(0.000) | 1.02(0.02) | 1.75(0.19) |
| | SSIM | 0.970(0.020) | 0.993(0.009) | 0.997(0.007) | 0.999(0.000) | 1.06(0.01) | -0.23(0.11) |
| StyleGAN2 | MSE | 0.945(0.035) | 0.989(0.011) | 0.990(0.020) | 0.987(0.011) | 1.13(0.16) | 1.32(0.34) |
| | SSIM | 0.945(0.047) | 0.987(0.015) | 0.988(0.027) | 0.991(0.008) | 1.08(0.18) | -0.17(0.26) |
| WaveGAN† | STFT | 0.529(0.229) | 0.892(0.088) | 0.661(0.313) | 0.938(0.062) | 1.09(0.05) | 4.67(0.30) |

0.995]. Measuring Hessian similarity using the statistics we developed $C^{Hlog}$ and $C^{Hlin}$ resulted in correlations concentrated at 0.99. Thus, we found that the Hessian matrices and their spectra were highly correlated across image distance metrics, and that the Hessian matrices had a similar effect on the eigenvectors of each other.

One major difference across Hessians from different image distance metric was evident in the scale of the eigenvalues. We regressed the log Hessian spectra induced by SSIM or MSE onto the log Hessian spectrum induced by LPIPS, and found the intercepts of the regression were usually not zeros (Tab. 5). This result showed different image distance metrics exhibit different "unit" or intrinsic scale, although they all factored out the same structure in the GAN.

This result is contextualized by Section 5. As equation 3 showed, the Riemannian metric or Hessian of the generator manifold is the inner product matrix of the Jacobian of the representational map. The effect of image distance metric on the Riemannian metric is to add a few more terms on top of the chain of Jacobians. The Jacobian terms from the layers of generator seem to have a larger effect than the final terms coming from the image distance metric.

Note that this does not mean that the choice of sample space distance function is irrelevant. Going beyond image generation, when applying our method to an audio generating GAN, the WaveGAN, we found that the choice of distance function in the space of sound waves will substantially affect the Hessian obtained. We used the MSE of wave forms and MSE of spectrograms (denoted by STFT) to compute metric tensor of that sound wave manifold. We found the element-wise Hessian correlation between these is around 0.53, while the other Hessian similarity metric are also lower than the counterparts for BigGAN, PGGAN and StyleGAN2 (Tab. 5). We think the MSE of spectrograms is a more perceptually relevant distance metric of sound waves than MSE of wave forms, and this difference is reflected in the geometry they induced i.e. anisotropy and homogeneity (Tab. 2, 4). Thus, when and how the sample space distance metric will affect the geometry of generative model still requires more development to be answered.

## A.6 RANDOM MIXING OF SPECTRA

Here we give a simple derivation of why a highly ill-conditioned Hessian matrix may appear normal, under the probe of random vectors. Given a symmetric matrix $H$, and its eigen decomposition $U\Lambda U^T$, we computed its effect on an isotropic random vector $v$, $\alpha(v) = v^T H v / v^T v$, and $v \sim \mathcal{N}(0, I)$. This random variable represents the effect of the symmetric matrix on random directions.

Note that a change of variable using the orthogonal matrix $\boldsymbol{w} = U^T \boldsymbol{v}$ will not change the distribution $\boldsymbol{w} \sim \mathcal{N}(0, I)$. Through this the random variable $\alpha$ could be rewritten as

$$\alpha(\boldsymbol{v}) = \frac{\boldsymbol{v}^T H \boldsymbol{v}}{\boldsymbol{v}^T \boldsymbol{v}} = \sum_i \lambda_i \frac{\|\boldsymbol{u}_i^T \boldsymbol{v}\|_2^2}{\|\boldsymbol{v}\|_2^2}, \ \boldsymbol{v} \sim \mathcal{N}(0, I) \tag{9}$$

$$= \sum_i \lambda_i w_i^2 / \sum_i w_i^2, \ w_i \sim \mathcal{N}(0, I) \ i = 1, 2 ... d \tag{10}$$

$$= \frac{1}{\sum_i w_i^2} \sum_i \lambda_i w_i^2 = \sum_i c_i \lambda_i \tag{11}$$

$$c_i := w_i^2 / \sum_i w_i^2 \tag{12}$$

As each element in $\boldsymbol{w}$ is distributed as i.i.d. unit normal, $w_i^2$ is distributed as i.i.d. chi-square distribution of parameter 1. $w_i^2 \sim \chi^2(1) \sim \Gamma(\frac{1}{2}, \frac{1}{2})$. Thus the normalized weights $c_i = w_i^2 / \sum_i w_i^2$ conform to a Dirichlet distribution $\boldsymbol{c} \sim Dirichlet(\frac{1}{2}, \frac{1}{2}, ... \frac{1}{2})$. Through moment formula of Dirichlet distribution, it is straightforward to compute the mean and variance of $\alpha(\boldsymbol{w}) = \boldsymbol{c}^T \lambda$

$$\mathbb{E}[\alpha] = \lambda^T E[\boldsymbol{c}] = \frac{1}{n} \sum_i \lambda_i \tag{13}$$

$$Var[\alpha] = \lambda^T Cov[\boldsymbol{c}] \lambda = \frac{2}{n(n+2)} \sum_i \lambda_i^2 - \frac{2}{n^2(n+2)} (\sum_i \lambda_i)^2 \tag{14}$$

$$Var[\lambda] = \frac{1}{n} \sum_i \lambda_i^2 - \frac{1}{n^2} (\sum_i \lambda_i)^2 = \frac{n+2}{2} Var[\alpha] \tag{15}$$

As we can see, the variance of the effect on random directions scales $1/n$ relative to the variance of original eigenvalue distribution. This is why the distribution is much tighter than the whole eigenvalue distribution.

This phenomenon may explain why the perceptual path length regularization (PPL) used in Karras et al. (2020) doesn't really result in a flat spectrum. In our notation, the PPL regularization minimizes $\mathbb{E}_{\boldsymbol{z}} \mathbb{E}_{\boldsymbol{v}} \|\alpha_{\boldsymbol{z}}(\boldsymbol{v}) - b\|^2$, which is to minimize the variance of the distribution of $\alpha(\boldsymbol{v})$ with $\boldsymbol{v}$ sampled from normal distribution and $\boldsymbol{z}$ sampled from latent distribution. The global minimizer for this regularization is indeed a mapping $G$ with flat spectrum, i.e. an isometry up to some scaling. However, we can see through our derivation and Fig.6 that even for highly anisotropic spectrum, this variance will not be very large. Thus we should expect a limited effect of this regularization.

### A.7 METHOD TO QUANTIFY METRIC SIMILARITY

We developed our own statistic to quantify the similarity of metric tensor between different points. Here we discuss the benefits and caveats of it.

Angles between eigenvectors *per se* are not used, since eigenvectors with close-by eigenvalues are likely to rotate into each other when computing eigendecomposition (Van der Sluis & Van der Vorst, 1987). However, the statistics should be invariant to this eigenvector mixing, and also take the eigenvalues into account. In our statistics, we applied the eigenvectors $U_1 = [\boldsymbol{u}_{1,i}, ...]$ of one matrix $H_1$ to the other $H_2$, i.e. examined the length of these vectors as measured by the other matrix as the metric tensor, $\alpha_{H_2}(\boldsymbol{u}_{1,i}) = \boldsymbol{u}_i^T H_2 \boldsymbol{u}_i$. Recall that $\alpha_{H_1}(\boldsymbol{u}_{1,i}) = \boldsymbol{u}_{1,i}^T H_1 \boldsymbol{u}_{1,i} = \lambda_{1,i}$ and $\alpha_{H_2}(\boldsymbol{u}_{2,i}) = \boldsymbol{u}_{2,i}^T H_2 \boldsymbol{u}_{2,i} = \lambda_{2,i}$. If the eigenvectors fall in the eigenspace of the same eigenvalue in $H_2$, then $\alpha(\boldsymbol{u}_{1,i})$ will equal the eigenvalue, and thus our statistic is invariant to rotation within the eigenspace. If the eigenvectors are totally uncorrelated, the resulting $\alpha(\boldsymbol{u}_{1,i})$ will distribute like that of random vectors as in Fig.6. As we compute the correlation between the eigenvalue $\lambda_{2,i} = \alpha_{H_2}(\boldsymbol{u}_{2,i})$ and the $\alpha_{H_2}(\boldsymbol{u}_{1,i})$, we summarize the similarity of action of $H_2$ on eigenvectors $U_1$ and $U_2$.

However, this method assumes an anisotropy of spectra in both metric tensors. For example, if both tensors are identity matrices $H_1 = H_2 = I$, then this correlation will yield NaN, as there is no variation in the spectra to be correlated. Similarly, if the metric tensor has a more isotropic spectrum,

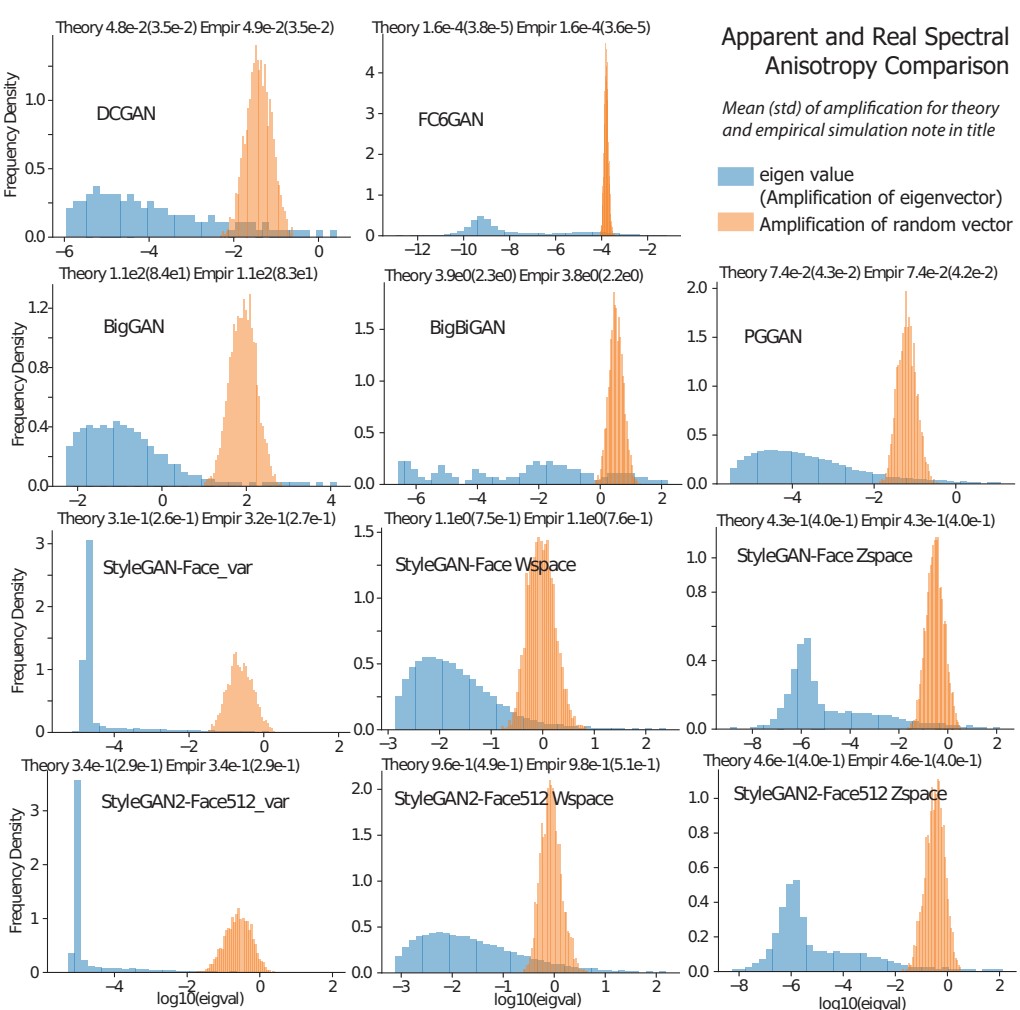

Figure 6: **Spectral Histogram compared to Apparent Anistropy in Different GAN models** (FC6GAN, DCGAN, BigGAN, BigBiGAN, PGGAN, StyleGAN, StyleGAN2) The apparent speed of image change $\alpha(\boldsymbol{v})$ has much smaller variability than the variability in the whole spectra. Eq. 13 can predict the mean and std of the apparent variability.

then it will generally have a smaller correlation with others. In that sense, spectral anisotropy also plays a role in our statistics for metric similarity or homogeneity of the manifold. In all the GAN spaces we examined, there is a strong anisotropy in the metric spectra, thus this correlation works fine. But there is caveat for comparing this correlation between two GANs when there is also difference in the anisotropy in their spectra, as a smaller anisotropy can also results in a smaller metric similarity.

Finally, we are aware that there are different ways to average symmetric positive definite matrices (SPSD), induced by different measures of distance in the space of SPSD(Yuan et al., 2020). Here we picked the simplest one to estimate the global Hessian in the latent space: averaging the metric tensors element by element.

## A.8    GEOMETRIC STRUCTURE OF WEIGHT-SHUFFLED GANS

Here we show the geometric analysis for the shuffled controls for all our GANs. Specifically, we shuffled the elements of the weight tensor from each layer to keep the overall weight distribution

Table 4: **Quantification of Manifold Homogeneity** by metric consistency $C_{ij}^H$ on log scale and linear scale. Same data generating Fig. 3 D. Models marked with † are audio wave form generating GANs.

| | Log scale | | Linear Scale | |
|---|---|---|---|---|
| | mean | std | mean | std |
| FC6GAN | 0.984 | 0.002 | 0.600 | 0.119 |
| DCGAN | 0.920 | 0.028 | 0.756 | 0.192 |
| BigGAN | 0.934 | 0.024 | 0.658 | 0.186 |
| BigBiGAN | 0.986 | 0.007 | 0.645 | 0.180 |
| PGGAN | 0.928 | 0.014 | 0.861 | 0.123 |
| StyleGAN-Face_Z | 0.638 | 0.069 | 0.376 | 0.160 |
| StyleGAN2-Face512_Z | 0.616 | 0.052 | 0.769 | 0.181 |
| StyleGAN2-Face256_Z | 0.732 | 0.037 | 0.802 | 0.130 |
| StyleGAN2-Cat256_Z | 0.700 | 0.040 | 0.689 | 0.151 |
| StyleGAN-Face_W | 0.878 | 0.037 | 0.780 | 0.190 |
| StyleGAN2-Face512_W | 0.891 | 0.048 | 0.838 | 0.127 |
| StyleGAN2-Face256_W | 0.869 | 0.052 | 0.756 | 0.159 |
| StyleGAN2-Cat256_W | 0.895 | 0.118 | 0.539 | 0.216 |
| WaveGAN_MSE† | 0.906 | 0.022 | 0.776 | 0.139 |
| WaveGAN_STFT† | 0.809 | 0.096 | 0.467 | 0.285 |

Table 5: **Hessian preconditioning improves GAN inversion** The mean and standard error of fitting score (minimum LPIPS distance to target using 4 random initial vectors) are presented in the table, which is the same data generating Fig. 5. For each GAN, target dataset pair, 200-300 different target images are used.

| GAN | Target Image | Hessian | | None | |
|---|---|---|---|---|---|
| | | Mean | SEM | Mean | SEM |
| StyleGAN1024 | CelebA | 0.334 | 0.002 | 0.344 | 0.002 |
| StyleGAN1024 | FFHQ | 0.231 | 0.003 | 0.236 | 0.003 |
| StyleGAN1024 | GANgen | 0.055 | 0.002 | 0.060 | 0.003 |
| PGGAN | CelebA | 0.179 | 0.003 | 0.207 | 0.003 |
| PGGAN | FFHQ | 0.175 | 0.003 | 0.197 | 0.003 |
| PGGAN | GANgen | 0.045 | 0.003 | 0.027 | 0.001 |
| BigGAN | ImageNet | 0.162 | 0.003 | 0.189 | 0.003 |
| BigGAN | GANgen | 0.144 | 0.003 | 0.195 | 0.003 |

unchanged. To show how learning affected the geometry of the image manifold, we computed the spectra and the associated metric consistency statistic for weight-shuffled GANs[2].

In Fig. 7, we showed that the shuffled controls exhibited flatter spectra and smaller top eigenvalues. There, the correlation of metric tensors in shuffled GANs shows an unclear result. In some GANs, there remains strong correlations in the metrics across locations, while in some, the correlation is close to zero. We think the reason is that our statistic for homogeneity (i.e. a correlation of action of metric tensors $C_{ij}^{Hlog}$) somewhat entangles homogeneity with the anisotropy of the space. That is, when the space has a totally flat spectrum (the map is isometric), then the correlation coefficient of action will be zero or nan, although the metric tensor will be the same everywhere. Thus the change of anisotropy and the change of homogeneity may interfere with each other, thus shuffling can result in a mixed result. We are working to develop new statistics that will measure the similarity of the Hessian, invariant of anisotropy.

---

[2]We were unable to obtain a sensible spectra for either shuffled or randomly initialized BigBiGAN possibly due to its architecture; but we show the comparison for all other models.

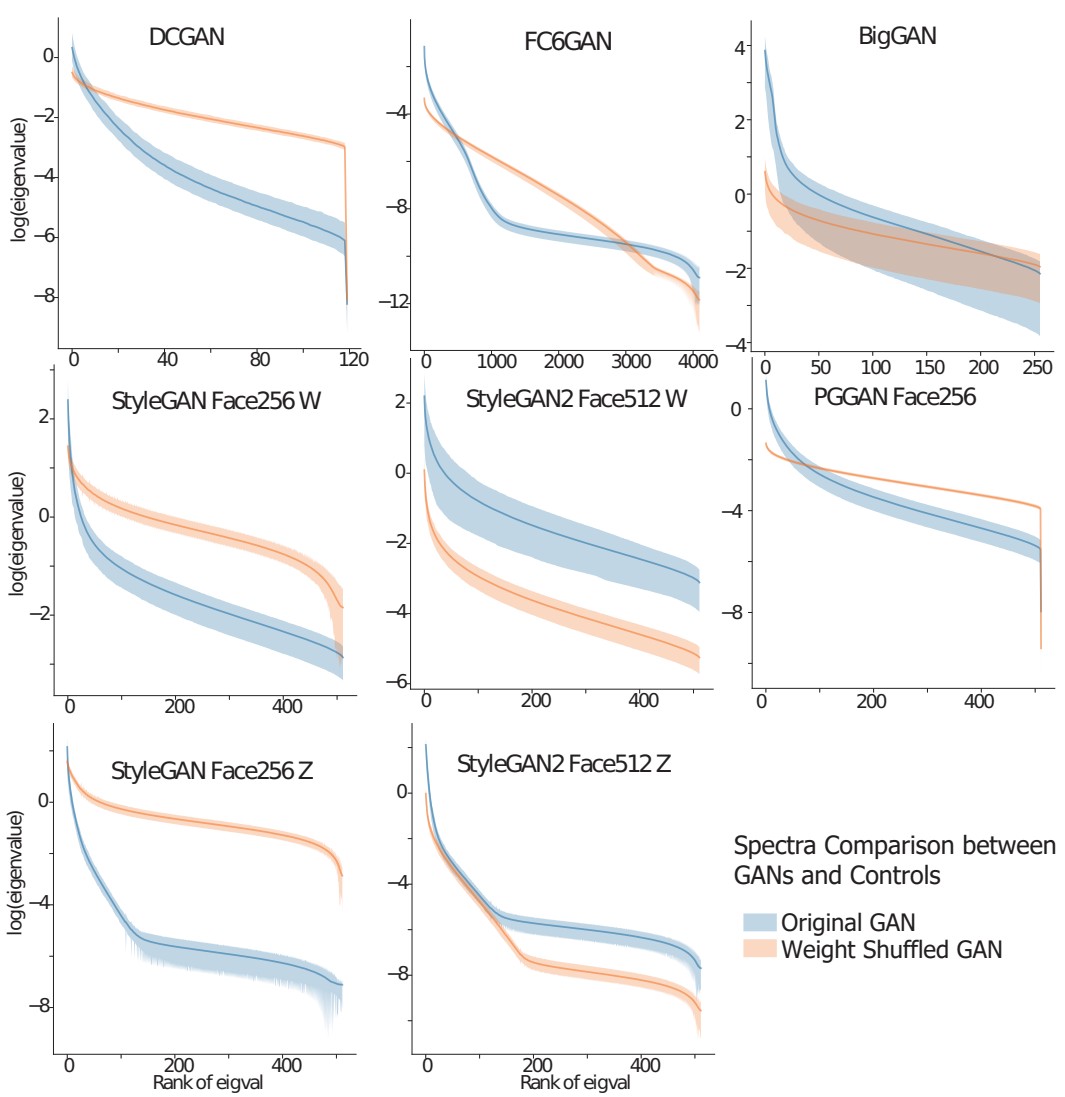

Figure 7: **Comparing Spectra of Original and Weight Shuffled GANs**: Most Shuffled GAN shows a slower spectral decay and a smaller maximum eigenvalue.

## A.9    Detailed Comparison to Previous Unsupervised Ways to Discover Interpretable Axes

Here we compare the axes discovered by our method with those from a previous approach. Specifically, we applied our method to the same pre-trained GANs used in Voynov & Babenko (2020), comparing the axes they discovered versus our Hessian structure. Although this method follows a quite different approach compared to ours and those of (Härkönen et al., 2020; Shen & Zhou, 2020), we thought it would be interesting to determine if the interpretable axes discovered in their approach had a relationship with the Hessian structure defined above. If so, this could serve as independent confirmation of the effectiveness of both types of approaches.

In their work, for each generative network $G$, two additional models were simultaneously trained to discover interpretable axes: a "deformator" $M$ and a "latent shift predictor" $P$. The "deformator" $M$ learned to propose vectors $\{v_i\}$ to alter the image, which were used to create images pairs $(G(z), G(z + v_i))$ using random reference vectors $z$; the "latent shift predictor" $P$ took in the images pairs and learned to classify the direction in which the latent code shifted $\hat{v}_i = P(G(z), G(z + v_i))$. The axes learned by the deformator $M$ were subsequently annotated and a subset was selected by humans.

Using their code, we compared these annotated axes with the Hessian structure we computed on their GANs (PGGAN512, BigGAN noise and StyleGAN Face). In PGGAN512, we found that their discovered axes had a significantly larger $v^T H v$ (i.e. approximate rate of image change) than random vectors in that space; in other words, their axes were significantly more aligned with the top eigen space ($P < 0.05$ for all axes). Further, we wanted to investigate whether their axes aligned with individual eigenvectors identified by our Hessian or whether their axes randomly mixed with our top eigen space. To achieve this goal, we search for the power coefficients that are significantly higher than expected from projection of unit random vector. In fact, for each and every of the discovered axes, we found 1-3 eigenvectors that they are significantly ($p < 5 \times 10^{-4}$) aligned to. Moreover, these strongly aligned eigenvectors are all in our top 60 eigen dimensions, in fact, 3 of their axes aligned with eigenvector 11 and 2 of their axes aligned with our eigenvector 6. (Fig.8 A) Moreover, we "purify" their axes by a) retaining projection coefficients only in top 60 eigenvectors, or b) retaining the coefficients only in the 1-3 strongly aligned eigen vectors and set all the other 500+ coefficients to zero, and compared their effect on a same set of reference vectors, using the same step size. We found that by project out coefficients in the lower space, the image transformation is perceptually very similar (Fig.8 B,C). If we only retains the eigenvectors that it highly aligns to, the image transform will be more different, but the annotated semantics in the transform seems to keep (Fig.8 D,E). Thus, their method also discovered that the top eigenspace of PGGAN contained informative transformations, and further confirmed that optimizing interpretability may improve alignment with individual eigen vectors rather than mixing all the eigen dimensions.

Note, as we project out coefficients, the resulting vector has a smaller than unit norm, thus we are moving a smaller distance in latent space using the same step size (Fig.8 B-E title). If we renormalize the vector to unit norm we will need to take a smaller step size to achieve the same transform. This is confirming our predictions in Sec. 4: Each interpretable axis $u$ has a family of equivalent axes $u + v$, which add a direction $v$ in the lower eigenspace or null space of the GAN. These axes encode the same transforms but the speeds of image change on them are different. In this sense, the top eigenspace could be used to provide a "purer" version of the interpretable axes discovered elsewhere.

Although both types of approaches are promising, by removing the need to train additional networks, our method can be viewed as a more efficient way to identify informative axes. Further work comparing axes discovered by different methods will elucidate the connection between interpretable axes and the Hessian structure more.

### A.10 DETAILED COMPARISON TO HESSIAN PENALTY

In this section, we compare our work with that of (Peebles et al., 2020), which also focused on the use of a Hessian matrix to "disentangle" directions in generator latent space. This *Hessian penalty* approach devised a stochastic finite difference estimator of the non-diagonal elements of the Hessian, using only a forward pass, resulting in an efficient regularization of Hessian diagonality during GAN training. This clever approach led to several benefits, including increases in smoothness in latent interpolation, and improvements in interpretability. However, there are clear contrasts between the Hessian penalty approach and ours.

At the highest conceptual level, both works relied on the analysis of the Hessian matrix. Our framework is motivated by a geometric picture of the image manifold and latent representations, while their work is motivated by the idea of increasing the independence of latent factors. The Hessian matrices involved are also different: we computed the Hessian of the squared distance metric $d^2$ in image space, while they they computed the Hessian of every pixel in the image.

At the implementation level, the methods are also different. We computed the exact Hessian matrices or their low-rank approximations based on backpropagation and the Lanczos iteration. They devised a stochastic finite-difference estimator of the total non-diagonal elements of the Hessian, using only a forward pass, which they termed the Hessian penalty. This remarkable achievement enabled them to efficiently regularize Hessian diagonality during GAN training. As another point of contrast, the Hessian penalty does not provide an explicit geometric picture as the exact Hessian matrices do in our work. Specifically, their penalty did not reveal the spectral structure of the Hessian matrices, which is encoded in the diagonal elements of the matrix. Because of this, the anisotropy can be explicitly demonstrated in our work.

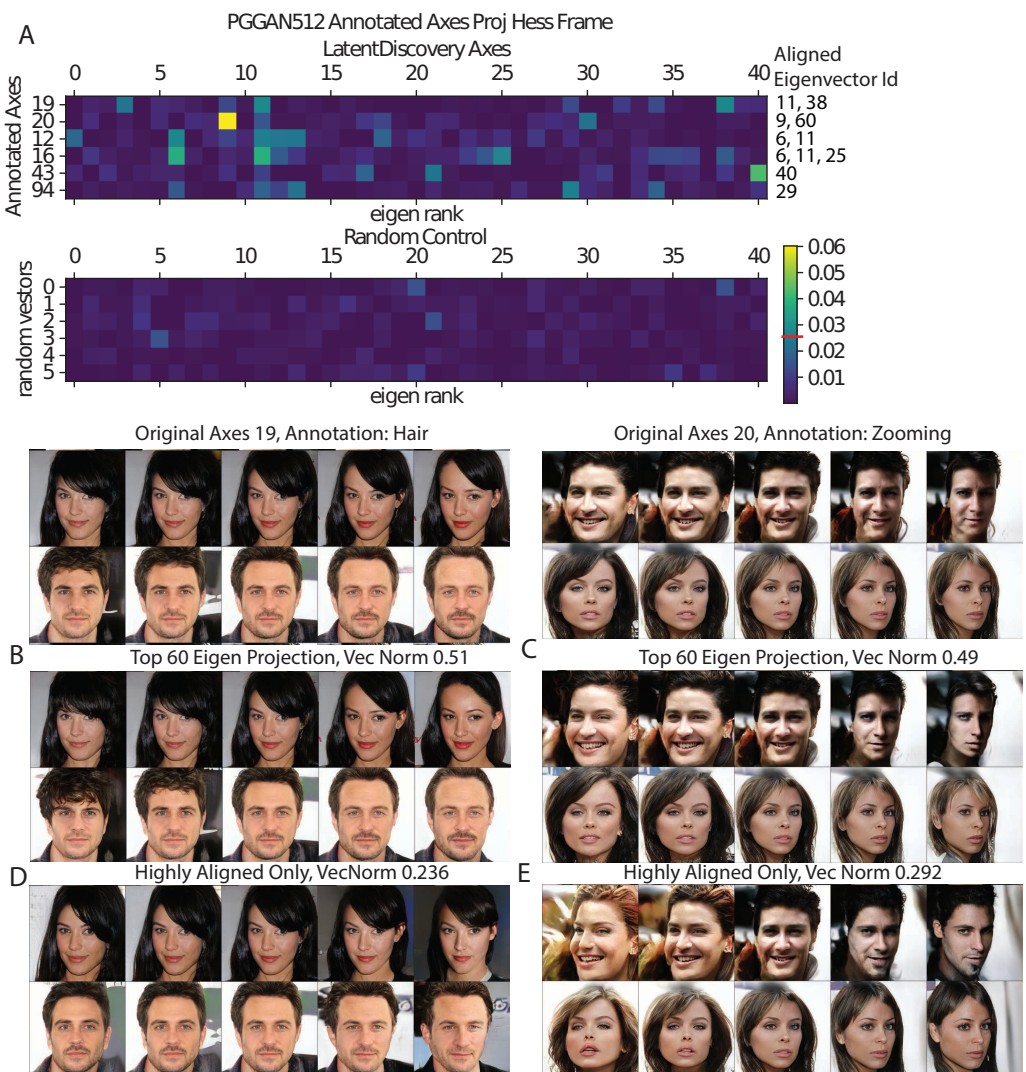

Figure 8: **Analyzing interpretable axes from (Voynov & Babenko, 2020) under the Hessian framework**. A. Projection power of their annotated interpretable axes and 6 unit norm random vectors on the top 40 eigen vectors. The color code is matched. The red line on colorbar denotes $p < 1 \times 10^{-4}$ threshold for the power value, and the significantly aligned axes $p < 5 \times 10^{-4}$ are indicated. B,C, Image transforms encoded by projection of 2 of their vectors (19, 20) into the top 60d eigenspace. D,E Similar to B,C, but their vectors are projected onto the aligned eigenvectors only. The norms of the projected vectors are noted on title. Panel B,D and C,E share the same reference image and the same step size across each mini column, though the distance travelled along BC and DE is smaller as the vector is shortened.

It is clear that the Hessian penalty provides a very important and complementary approach to ours. This work showed that regularizing Hessian diagonality during training could promote disentangled latent representation using PGGAN (this provides a more detailed evaluations of disentanglement than we achieved).

Interestingly, many of the phenomena they observed in Hessian-penalized generative networks were reminiscent to phenomena we observed in normally trained generative networks with a Hessian eigen-frame rotation of the latent space. As a first example, they found that after applying the Hessian penalty, many factors stopped affecting the image (termed "latent space shrinkage" in their work). We also found this phenomenon in normal pre-trained GANs, i.e. they showed a large bottom eigenspace with close to 0 eigenvalues, in which the eigenvectors generated small to no changes in

the image (termed "null space" in our work; Sec. 4). Thus if we performed a Hessian eigen-frame rotation, non-Hessian-penalized generators will exhibit similar behavior as theirs do.

As a second example, they also found that enforcing the Hessian penalty on middle layers is helpful in regularizing the Hessian diagonality in the image space, which is reminiscent of our finding that the Hessian eigen-frames are usually well aligned across the layers of generator (Sec. 5), though the spectra get shaped throughout the layers.

The Hessian penalty raises an interesting question: what makes a diagonal Hessian matrix special? Because the Hessian is a real symmetric matrix, at each point $z$, it can be diagonalized by a rotation of the latent space. However, to achieve a diagonal matrix across the latent space, Peebles et al. (2020) had to (implicitly) enforce each point to share the same rotation (i.e. Hessian eigen-frame). Given the training involved, this encouraged the homogeneity or flatness of latent space as identified in our framework. However, as shown in our work, most GANs exhibit homogeneity or flatness in their latent space even without the Hessian penalty. So it would be interesting to compare generators trained with Hessian penalty against those trained without the penalty but with a *post hoc* rotation of the latent space using the global eigen-frame. We expect that their training will exhibit flatter geometry than the *post hoc* rotated latent space. However, even if this is not true, it would still be interesting if this flat geometry can emerge from modern GAN training i.e. fitting the natural image distribution.

Finally, aside from regularizing GAN training, Peebles et al. (2020) also explored the use of the Hessian penalty in finding interpretable directions in pre-trained BigGAN (BigGAN-128). Similar to Voynov & Babenko (2020), the axes they discovered showed a striking correspondence to those identified in our approach (Fig. 9). We showed that when we computed the Hessian at a few points in the noise space of the generator, the interpretable axes they found aligned well with our top eigenvectors, with a one-to-one or one-to-two matching. For example, their reported interpretable axes 5, 6 and 8 (for the golden retriever class) had 0.998, 0.964, 0.990 of their power concentrated in our top 10-dimensional eigenspace. Moreover, they aligned with single eigenvectors 0, 5 and 2 with power 0.988, 0.499, 0.852 respectively. Due to the presence of close by eigenvalue, eigen vectors can mix into each other, resulting in the phenomenon that the axes identified in Peebles et al. (2020) can correspond to a few adjacent eigenvectors (e.g. their axes 5 correspond to our eigenvectors 4 and 5, with corresponding eigenvalues 8.3 and 7.4; as reference, eigenvalue 3 = 17.6, eigenvalue 6 = 5.6, a much larger gap.). We found that power concentration within the top eigenspace is a great indicator of the interpretability of their reported axes: all but one of the reported axes showed over 0.95 power concentrated in top 10d eigenspace, while the axes they found not to be interpretable showed power $0.103 \pm 0.141$ (mean+-std) in the top 10d eigenspace. One advantage of our approach is computational efficiency. As the alternative method required learning (i.e. iterative optimization of a mixing matrix), it took us 40 minutes$\times$50 epochs to finish the training on a 6GB GTX 1060 GPU. In contrast, the present method directly computes the Hessian matrices at one- or a few points, taking 12 seconds to compute a full Hessian matrix (5-50 points usually suffice). The reason for this difference is that their method optimizes for a basis that diagonalizes the Hessian matrix based on a noisy estimate of diagonality – the Hessian penalty; in contrast, our approach directly computes and diagonalizes the matrix at given points. Finally, our method orders the axes by the eigenvalues, facilitating focus on the top eigenspace, and thus alleviating the need to go through all the axes to find interpretable ones.

In summary, we believe that as a stochastic regularizer of the Hessian matrix, the Hessian penalty is a valuable and complementary approach to ours. Our methods provide additional value by accurately estimating the top eigenvectors and eigenspectrum, suitable for analyzing geometry *post hoc*. However, unlike the Hessian penalty, direct use of our method to regularize training may be inefficient. It would be interesting to explore whether there is a middle ground that incorporates the advantages of both their estimator and our more precise calculation.

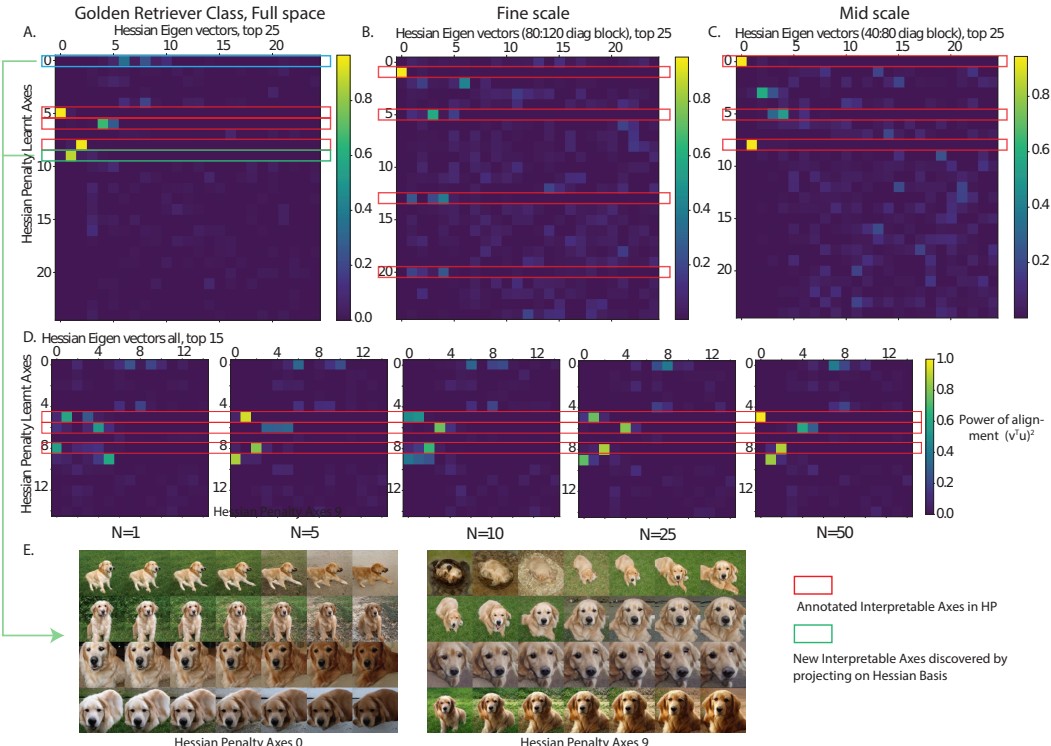

Figure 9: **Analyzing interpretable axes from (Peebles et al., 2020) under the Hessian framework**. A. Projection power of Hessian-penalty-identified axes on the top 25 eigen vectors. B,C. For the axes corresponding to fine or mid-scale changes, we performed eigen decomposition to the corresponding 40d diagonal block of the averaged full Hessian. We then computed the projection power with their axes. D. Comparison of the alignment with the Hessian eigen frame averaged over different number of points ($N = 1, 5, 10, 25, 50, 100$) in the full space. Although the specific matched eigen id varies between $N$, those interpretable axes all remain in our top eigenspace, with power over 0.95. E. Visualize the axes in Peebles et al. (2020) with a good alignment with our top 10d eigenspace (axes 0 and 9, boxes marked in green), other than those annotated in their paper (boxes marked in red). They are arguably interpretable as well.

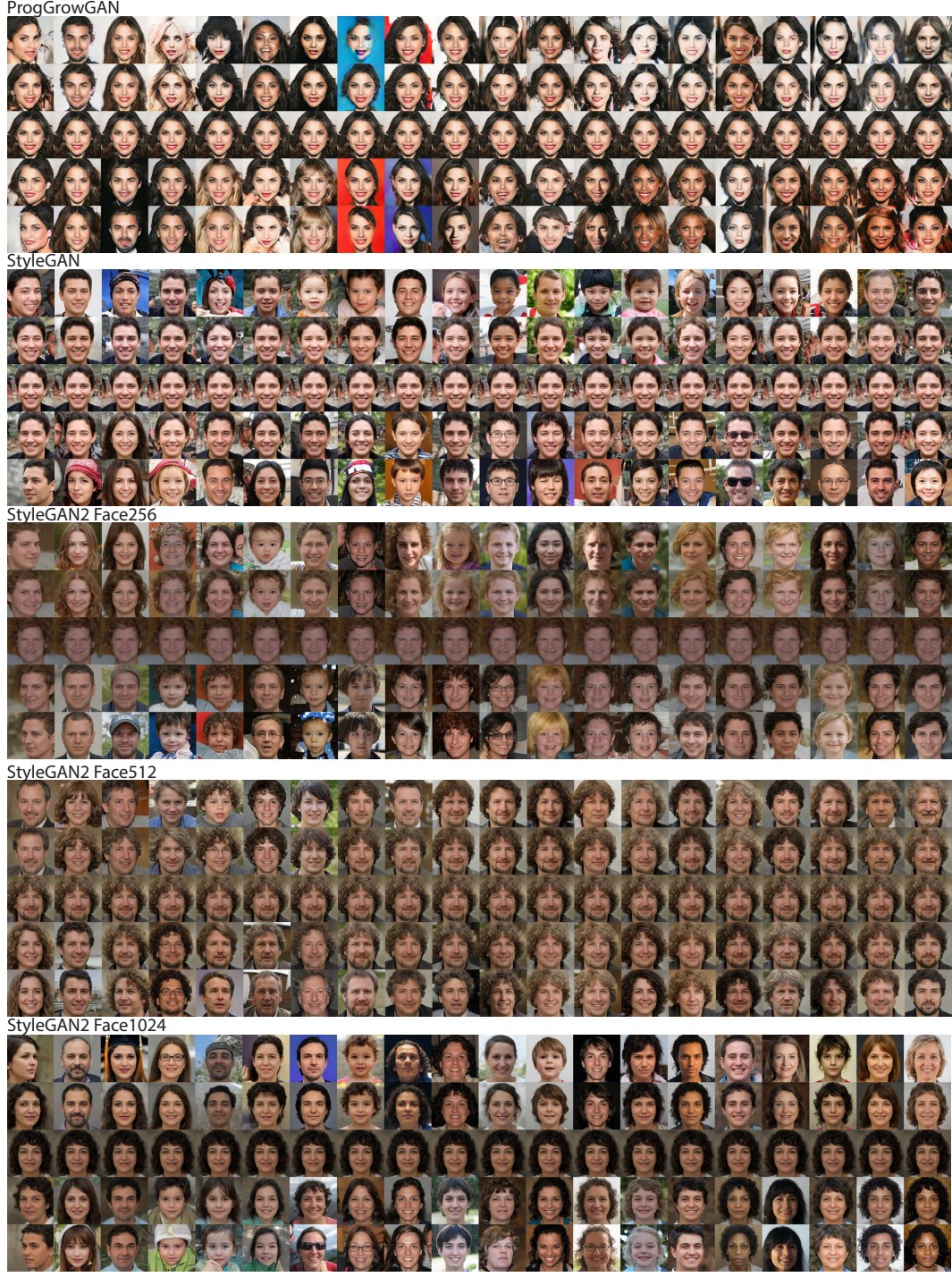

Figure 10: **Similar transforms encoded in the top eigendimension of GANs trained on face dataset**. Linear exploration along top 20 eigenvectors from origin in latent space are showed for each GAN. Linear equi-distance sampling on each eigenvector occupies a column and their eigenvalues are sorted in descending order from left to right. Step size along each vector is adjusted according to its eigenvalue for best continuity.

StyleGAN2 Cat Eig1          StyleGAN Face Eig1

StyleGAN2 Cat Eig4          StyleGAN Face Eig3

StyleGAN2 Cat Eig10         StyleGAN Face Eig5

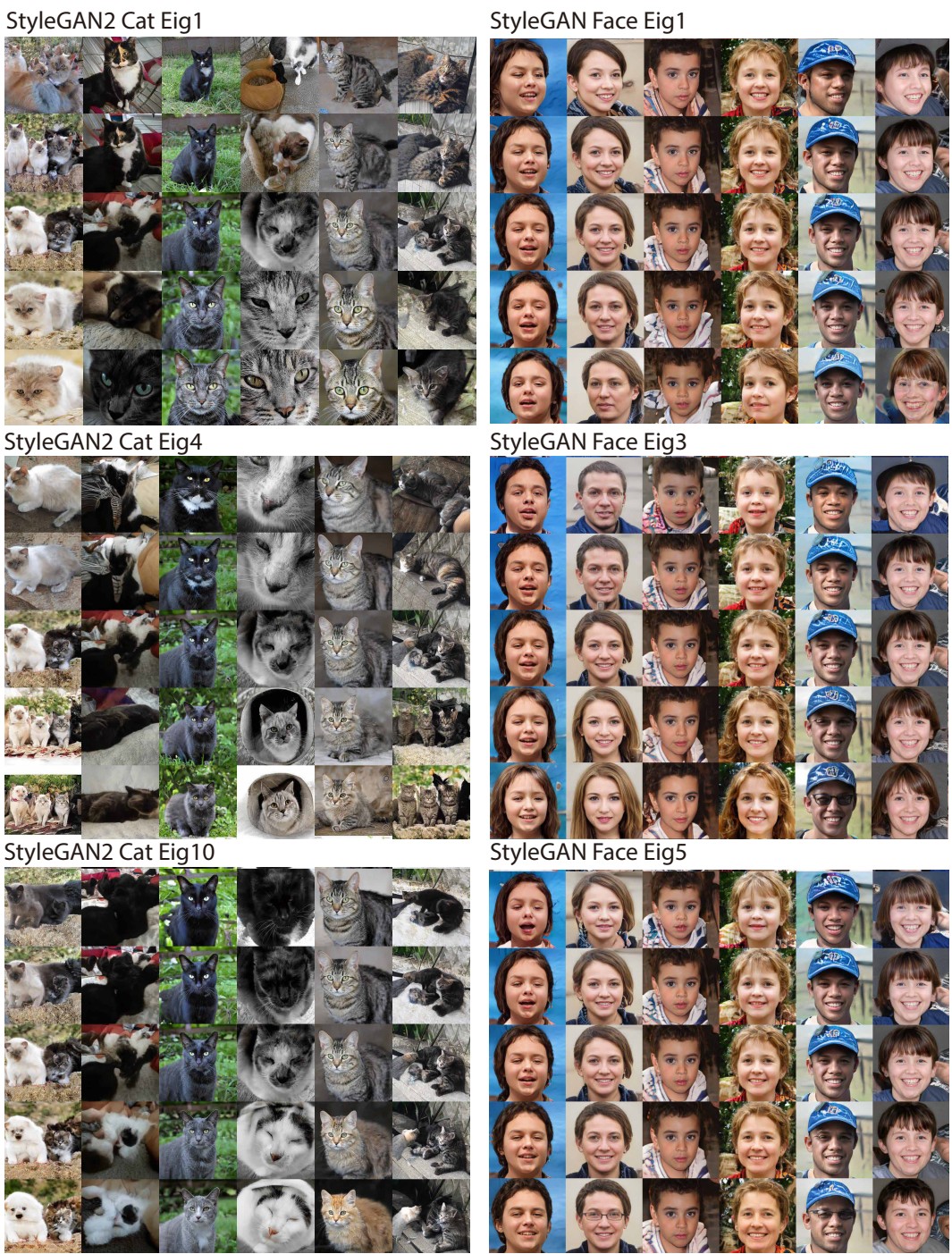

Figure 11: **Top eigenvectors encode similar transforms around different reference images**. Linear equidistant explorations from six randomly chosen reference images along the eigenvectors of averaged Hessians. These show qualitatively similar transforms to images — for example, proximity of Cat face (Eig1), proximity and cat number (Eig4), fur color darkening (Eig10) in StyleGAN2 Cat; face direction (Eig1), masculine vs feminine (Eig3), child vs adult (Eig5) in StyleGAN Face.

