# OpenReview forum: "A Geometric Analysis of Deep Generative Image Models and Its Applications"
_ICLR.cc/2021/Conference — ICLR 2021 Poster_

### Official Review · AnonReviewer1 · 2020-10-28
**Official Blind Review #1**

**Rating:** 5
**Confidence:** 4

**Review:**

_Summary_:
This work intends to explore the geometry of the latent space and proposes to define the distance in latent space as the distance between the corresponding generated images and use the Hessian of that squared distance as metric tensor to define the manifold. Using Learned Perceptual Image Patch Similarity (LPIPS), they show that the Hessian can either be computed through backpropagation or if that is not efficient, it is sufficient to iteratively compute the eigenvectors corresponding to the largest eigenvalues. With the proposed method, the empirical observations showed 1) the impact of those eigenvectors through examples, 2) consistent geometric local changes over different positions in the latent space, and 3) the impact of top eigenvectors on particular layers. Further, the authors discuss three areas of possible application (gradient-based GAN inversion, gradient-free image search, interpretable axes discovery).

_Strengths_:
- The paper addresses an interesting topic (understanding the latent space of GANs) and proposes a straightforward method.
- The empirical observations demonstrates the advantages of using an image similarity metric as latent space distance and the correlation between eigenvalues corresponding to the largest eigenvalues and image perception.

_Weaknesses_:
- The method is depending heavily on the distance metric employed, however, does not discuss in what way this could influence the outcome. In this paper, only LPIPS was considered as metric. What are the advantages and disadvantages? How would the analysis and outcome change when other distance metrics are used? These are questions that might benefit the work to discuss.
- The actual method was barely discussed in the paper, but rather moved to the appendix. I think the authors may want to restructure the paper to include how to compute the eigenvectors are computed.
- The actual application of the proposed method was rather discussed than actually performed. I do think this is a very interesting proposal, however, with only showing empirical observations, I believe the scope of this paper is too little. It would have been better if the authors picked one of the applications and applied their method.

For more details see below.

_Overall assessment_:
Overall, I find the proposed method very interesting and the empirical observations compelling. However, I find the scope of the paper not sufficient as it would have been nice to see that one of the application with the usage of the Hessian metric as distance function would have worked.

_Detailed questions and comments_:
- Learned Perceptual Image Patch Similarity (LLIPS, Section 3 "Numerical Method"): As this is the main distance metric being used, would it be possible to briefly introduce and define it in the paper?
- Requirement for understanding the latent space (abstract): The abstract mentions inversion and interpretability as requirement for understanding the latent space. It was not clear to me how the method presented is addressing each of the requirement. Further, the abstract also claims "This geometric understanding unifies previous results of GAN inversion and interpretation.". Can you clarify these claims?
- Appendix A.2 Methods for computing the hessian: Parts of how the Hessian is computed should be in the main paper as this is the main method for this paper. The explanation in Section 3 "Numerical Method" was superficial and I could have not understood the methodology without reading the the appendix.
- Spectrum Structure of GANs (Section 4, Figure 2): Can you clarify what has been exactly used to plot Figure 2? In the paragraph, Figure 2 was plotted using the mean and 90% confidence interval, Figure 2 y-axis label says $\log(eig/eigmax)$.
- Figure 3D: The figure is hard to read, with many data points overlapping each other. I am also confused what the two lines crossing each data point represents.
- "Then we explored linearly the latent space along the eigenvector" (Section 4): Why is the exploration linearly? Does this conform with the manifold? How is $\mu_i$ defined? The footnote also says that spherical linear exploration is used some spaces. Can you elaborate more on how you performed exploration?
- Top Eigenvectors Capture Perceptual Relevant Changes: Would it be able to quantify this? As only four samples were shown, how would we know that this generalizes for all samples?
- [1]-[4] might be also relevant to the topic of latent space exploration. They are not necessarily w.r.t. to manifold learning, however, with respect to applications in Section 6, they look very close.

_Post-rebuttal_:
I do highly appreciate the authors trying to answer all our questions and adding more details and experiments. However, after also reading through [Chiu et al.,  SIGGRAPH 2020] I do find that this paper has a large overlap with the one mentioned. Therefore, agreeing with Reviewer #2, the contribution is reduced to applying the idea to GANs. Therefore I am keeping my recommendation.

[1] Lipton, Z.C. and Tripathi, S., 2017. Precise Recovery of Latent Vectors from Generative Adversarial Networks. ICLR 2017 workshop.
[2] Albright, M. and McCloskey, S., 2019, May. Source Generator Attribution via Inversion. In CVPR Workshops (Vol. 7).
[3] Webster, R., Rabin, J., Simon, L. and Jurie, F., 2019. Detecting overfitting of deep generative networks via latent recovery. In Proceedings of the IEEE Conference on Computer Vision and Pattern Recognition (pp. 11273-11282).
[4] Bojanowski, P., Joulin, A., Lopez-Pas, D. and Szlam, A., 2018, July. Optimizing the Latent Space of Generative Networks. In International Conference on Machine Learning (pp. 600-609).

---

> ### Author Response · Authors · 2020-11-22
> **Response to AnonReviewer1: Our result is reproducible using LPIPS, MSE and SSIM. (1/3)**
>
> We thank our reviewers for their time in considering our work. These insights have improved our paper considerably. We respond to each point below.
>
> > The method is depending heavily on the distance metric employed, however, does not discuss in what way this could influence the outcome. What are the advantages and disadvantages? How would the analysis and outcome change when other distance metrics are used? These are questions that might benefit the work to discuss.
>
> The reviewer is correct that our work used LPIPS distance as the primary image dissimilarity metric, and thus, we set out to determine if our results were entirely dependent on this choice. We performed a set of additional experiments computing the metric tensor at the same hidden vector using different image distance functions, including the mean squared error in pixel space (MSE) and structural similarity index measure (SSIM). We computed the Hessian using each of these three measures (MSE, SSIM, and LPIPS) at 100 random sampled vectors in BigGAN, Progressive Growing GAN (Face), StyleGAN2 (FFHQ 256 resolution). We then compared the Hessians obtained with each image distance function. We found that the element-wise Pearson correlation of the Hessian matrices ranged between 0.94-0.99, the correlation of eigenvalue spectra ranged between 0.987-0.995. Using our derived statistics, we also measured the Hessian similarity  $C^{Hlog}$ and $C^{Hlin}$ and found this resulted in a similarly high correlation (~0.99). Thus we can confirm that the estimated Hessian matrix and its spectra are not strongly dependent on the chosen image distance metric, and their effect on the eigenvector of each other is correlated. We have added the table quantifying this result to the appendix.
>
> This result can be interpreted in the context of Section 5 of the manuscript. As the equation $H(z)=J_{\varphi\circ G}^TJ_{\varphi\circ G}$ there showed, the Riemannian metric of the GAN is the inner product matrix of the Jacobian of the generator composing the representation map for the distance metric $J_{\varphi\circ G}$. This Jacobian is the composition of a chain of Jacobian of each layer, and the effect of the image difference metric is to add a few more terms to the top of the chain of Jacobians. In this regard, the Jacobian terms from the layers of the GAN seem to have a dominating effect compared to the few final terms coming from the image dissimilarity metric.
>
> Another explanation is that the 3 metric functions are in consensus for the geometry on the GAN manifold, but they may not agree with each other for **out of manifold transforms**. For example, as the demo in [LPIPS](https://github.com/richzhang/PerceptualSimilarity) paper showed, for some transforms like image blur or color distortion, MSE, SSIM and LPIPS can yield different answers about which image is more similar. But image blur and color distortion are artifacts that are easily discriminated against, so I believe generators will be trained not to generate blur and color distortion. In this regard, these transforms are out of manifold transforms. Thus, even if LPIPS, SSIM, and MSE are not always the same,  they could provide a similar measure for the "on manifold transforms" encoded by GAN.
>
> > The actual method was barely discussed in the paper, but rather moved to the appendix. I think the authors may want to restructure the paper to include how to compute the eigenvectors are computed.
>
> We appreciate the reviewer's effort to dig into the appendix and understand our method. We agree with the reviewer: although originally we were compelled to move the detailed version of the numerical method to the Appendix due to page limits, now we have expanded the method part in the main text to make it more intuitive.
>
> > The actual application of the proposed method was rather discussed than actually performed. I do think this is a very interesting proposal, however, with only showing empirical observations, I believe the scope of this paper is too little. It would have been better if the authors picked one of the applications and applied their method.
>
> This is a good recommendation. In this initial study, our goal was to emphasize the multiplicity of applications that this observation can advance. To clarify, we performed experiments in Section 6 and showed that using Hessian can actually assist optimization in GAN space with or without gradient. In this revision, figure 5 is expanded to illustrate this result. However, we take the point that it would be desirable to really focus on one of the applications, and we are pursuing similar efforts. For this manuscript revision, we have chosen to emphasize GAN inversion: we spend more effort to show that the Hessian preconditioning technique will improve GAN inversion for multiple GANs (StyleGAN face, PGGAN) for in distribution and out of distribution samples, using multi-start Adam optimizer. Hopefully, this can expand the scope of the paper.

---

> ### Author Response · Authors · 2020-11-23
> **Response to AnonReviewer1: Explanation to the Detail Concerns. (2/3)**
>
> > Learned Perceptual Image Patch Similarity (LPIPS, Section 3 "Numerical Method"): As this is the main distance metric being used, would it be possible to briefly introduce and define it in the paper?
>
> LPIPS is a commonly used image distance metric based on pre-trained CNNs (e.g. VGG, AlexNet or SqueezeNet): it learns linear weights to combine the feature maps in order to match human perceptual judgments of similarity.
> We have now added a brief introduction for this metric in the Methods section.
>
> > Requirement for understanding the latent space (abstract): The abstract mentions inversion and interpretability as requirement for understanding the latent space. It was not clear to me how the method presented is addressing each of the requirement. Further, the abstract also claims "This geometric understanding unifies previous results of GAN inversion and interpretation.". Can you clarify these claims?
>
> These are fair concerns, which have prompted us to rewrite our Abstract to be more precise and less overreaching. First, regarding the issue of how our method addresses both inversion and interpretability (two criteria we believe are essential to understanding latent spaces), we address this primarily in Section 6 (Application); this section shows how the geometric knowledge gained from computing the Hessian improves GAN inversion and assist in the discovery of interpretable axes (although we admit this last part requires a more detailed comparison and quantification, as noted above).
>
> Second, regarding the issue of unification, we now make a more qualified statement: although we believe that this geometric framework unifies previous results on the **unsupervised discovery of interpretable axes** in the GAN (as discussed at the end of Section 5), it is not accurate to say that it unifies previous results on **GAN inversion**. Here is what we mean: previous work had described three ways to identify informative axes in GAN latent space: a) performing singular value decomposition (SVD) on initial-layer weight matrices of the GAN \citet{shen2020GANSemFact}, b) performing principal component analyses on middle-layer activations of the GAN  \citet{harkonen2020ganspace}, c) simultaneously train an axes proposer and a decoder which could identify a proposed axes when image moves in that direction \cite{voynov2020unsupInterpDir}. In this paper, we provide a larger context that unifies these approaches (Section 5). We find that a given GAN’s eigenstructure is built up throughout layers, and the eigenstructures of the intermediate layers are highly correlated with the eigenstructure of the full generator. Thus the top singular vectors as found in \citet{shen2020GANSemFact} corresponded to the top Hessian eigenvectors of the generator. The principal components of middle layer proposed in \citet{harkonen2020ganspace}, roughly correspond to the left singular vectors, and numerical experiments showed these PC axes concentrate in the top eigenspace of the Hessian. As for \cite{voynov2020unsupInterpDir}, we compared the axes they annotated and our Hessian, and found their axes concentrate more in our top eigen space. We interpret this agreement in the following way, the axes in the top space will create more obvious image change with a small step, thus make the decoding easier. We add more numerical comparison in the appendix to prove this point.
>
> > Appendix A.2 Methods for computing the hessian: Parts of how the Hessian is computed should be in the main paper as this is the main method for this paper. The explanation in Section 3 "Numerical Method" was superficial and I could have not understood the methodology without reading the appendix.
>
> We agree, and have expanded the method parts of the paper.
>
> > Spectrum Structure of GANs (Section 4, Figure 2): Can you clarify what has been exactly used to plot Figure 2? In the paragraph, Figure 2 was plotted using the mean and 90% confidence interval, Figure 2 y-axis label says log⁡(eig/eigmax)
>
> For Figure 2, we computed the Hessian at $k$ randomly sampled vectors, did an eigendecomposition of it, and collected the $k$ spectra. The solid line represents the average of the $k$ spectra, and the shading represents the 5th-95th percentile of each eigenvalue $\lambda_i$ across $k$ vectors (these are plotted on the log scale). Finally, as different GANs have different top eigenvalues, for the sake of comparison, we normalized the max eigenvalue of the average spectrum to be 1 for each GAN. We have re-written the Figure legend to clarify these points..

---

> ### Author Response · Authors · 2020-11-23
> **Response to AnonReviewer1: Explanation to the Detail Concerns. (3/3)**
>
>
> > Figure 3D: The figure is hard to read, with many data points overlapping each other. I am also confused what the two lines crossing each data point represents.
>
> We have now re-drafted this figure to improve its clarity; as the consistency measure $C_{log}$ and $C_{lin}$ is a pairwise measure, the crossing line through each dot signifies the standard deviation of the statistics across all pairs of points ($n(n-1)/2$ pairs). It shows the variability of the consistency measure within each GAN space. The data used to plot this figure is presented in Table 4. Quantification of Manifold Homogeneity.
>
> > "Then we explored linearly the latent space along the eigenvector" (Section 4): Why is the exploration linearly? Does this conform with the manifold? How is defined? The footnote also says that spherical linear exploration is used some spaces. Can you elaborate more on how you performed exploration?
>
> Conceptually speaking, this point is really interesting. “Linearly” refers to the path the latent vector traverses in the latent space - a straight line $z(s)=z_0+sv$, as $v$ is the tangent vector. However, as the GAN is a nonlinear mapping, the generated image sequence $G(z(s))$ does not traverse a straight line in the image space; thus in image space, the trajectory occurs on a manifold defined by $G$.
>
> For the radial exploration, this is inspired by the work in [2016White] and by some of our observations. It is defined by the following equation:
> 	$z(s)=\cos(s) z_0 + \sin(s)\bar v *\|z_0\| / \|\bar v\|$
> Geometrically, this curve is the geodesic on the sphere $S^{n-1}$ with the starting point $z_0$ and tangent vector $v$. We justified this by the fact that during training, as the latent vector is sampled from a Gaussian distribution, these points statistically concentrate on a very thin spherical shell ($\|z\|\sim \sqrt n$). Any deviation from this shell may or may not generate natural samples depending on the GAN. Mathematically, exploration along a straight line is going to change the norm, thus we use this radial exploration scheme. We chose between linear and spherical explorations based on which generated more natural samples for a given GAN.
>
> > Top Eigenvectors Capture Perceptual Relevant Changes: Would it be able to quantify this? As only four samples were shown, how would we know that this generalizes for all samples?
>
> This is an important point also raised by other reviewers. Mathematically, as the Hessian is correlated across sampled points, we remain certain that travelling along the top eigenvectors will generate fast and obvious changes to the image. We can numerically show this by comparing the Hessian eigenvalues and the real image dissimilarity curve, which we added to the appendix.
>
> However, as perceptual relevance is in the eye of the beholder, we cannot prove its generalizability by numerical experiments. In response, we are conducting a rapid human perceptual study on MTurk to quantify the interpretability of these top Eigenvectors. In short we operationalize the concept of interpretability by 3 criterions: the change is obvious, the change is consistent across different reference images and that the observers report it’s easy to interpret the change. We will append the result to the manuscript as we have it, and qualify any major changes to our results.
>
> > [1]-[4] might be also relevant to the topic of latent space exploration. They are not necessarily w.r.t. to manifold learning, however, with respect to applications in Section 6, they look very close.
>
> We appreciate the reviewer for pointing out these related works!

---

### Official Review · AnonReviewer4 · 2020-10-28

**Rating:** 6
**Confidence:** 4

**Review:**

The paper performs the analysis of the GAN latent spaces from the geometric perspective, inducing a metric tensor in the latent space from the LPIPS distance in the image space. The main authors' finding is that under such metric, the latent spaces of typical GANs are highly anisotropic, which can be exploited for more effective GAN inversion. Furthermore, the authors show that eigen vectors of the metric tensor often correspond to interpretable latent transformations.

Pros:

1) The paper is exceptionally well-written and provides a very interesting read. While the performed analysis is simple and natural, it does reveal several interesting findings about typical latent spaces: LPIPS-anisotropy, global consistency of the metric tensor.

2) The authors confirm the usefulness of their analysis by providing immediate practical benefits: more effective GAN inversion, which accounts for the latent space anisotropy.

Cons:

1) Missing work on interpretable GAN directions:

[A] The Hessian Penalty: A Weak Prior for Unsupervised Disentanglement, ECCV 2020

2) In my opinion, the authors do not provide enough support for their claim "This finding unifies previous unsupervised methods that discover interpretable axes in the GAN space".

- While the proposed method does seem to generalize both Ha ̈rko ̈nen et al., 2020 and Shen & Zhou, 2020, I do not see, how it captures Voynov & Babenko, 2020 and Pebbles et al (see above [A]). Furthermore, I believe that such claims should be supported by the experiments.  Could the authors experimentally confirm that their method results in the same set of directions as the existing methods?

Overall, I am positive about this submission, since the main analysis is both interesting and practically useful. My main criticism is that in terms of discovery of interpretable directions, the methods should be experimentally compared to existing alternatives. If it does provide a super-set of directions, obtained by existing methods, this would make the submission much stronger. Otherwise, the claim about unification should be toned down in my opinion.

======== AFTER REBUTTAL ========

I appreciate the authors' efforts on additional thorough comparison to existing works on interpretable axes discovery. From the updated manuscript, however, it is not clear what method is superior and the authors' approach appears to be a yet another method for this task rather than generalization of previous ones. Overall, I am still on the positive side since the observed findings deliver a clear profit for GAN inversion. But I am not increasing my score given that the "interpretable axes" part has become less impressive (in terms of weaker claims and conclusions) and the competing SIGGRAPH work.

---

> ### Author Response · Authors · 2020-11-24
> **Respond to AnonReviewer4: Comparison with Voynov & Babenko, 2020 Added to the Appendix**
>
> > *Missing work on interpretable GAN directions:**[A] The Hessian Penalty: A Weak Prior for Unsupervised Disentanglement, ECCV 2020*
>
> Thank you for bringing this work to our attention. We have now included it as a reference and as a launching pad for our investigations.
>
> > *In my opinion, the authors do not provide enough support for their claim "This finding unifies previous unsupervised methods that discover interpretable axes in the GAN space".**While the proposed method does seem to generalize both Ha ̈rko ̈nen et al., 2020 and Shen & Zhou, 2020, I do not see, how it captures Voynov & Babenko, 2020 and Pebbles et al (see above [A]). Furthermore, I believe that such claims should be supported by the experiments. Could the authors experimentally confirm that their method results in the same set of directions as the existing methods?*
> >
> > *Overall, I am positive about this submission, since the main analysis is both interesting and practically useful. My main criticism is that in terms of discovery of interpretable directions, the methods should be experimentally compared to existing alternatives. If it does provide a super-set of directions, obtained by existing methods, this would make the submission much stronger. Otherwise, the claim about unification should be toned down in my opinion.*
>
> We appreciate these kind comments! We agree that our submitted draft needed some clarification regarding the claim that our results unified previous unsupervised methods. To address this, we performed additional comparison with previous work on this topic, and have added those to the appendix.
>
> Specifically, we acknowledge that the *Voynov & Babenko, 2020* result may not be deducible from our result. To clarify the relationship, we performed an additional comparison of the axes they discovered and the Hessian eigenvectors we computed on those GANs. In the PGGAN, we found that the 6 axes they annotated have a larger alignment with our top eigenspace, as the $vHv$ for their axes are significantly larger than those of unit random vectors ($p<0.01$ for all but one axes $p<0.05$). Further, we projected their axes onto the eigenvectors of Hessian and analyzed how the power (square of projection coefficient) is distributed on the spectra. Further, to investigate the alignment of their axes and single eigenvector, we project their axes onto the basis formed by Hessian eigenvectors and compute the power (squared projection coefficient). We computed the histogram of power in different parts of spectra and found that the entropy of this distribution is significantly lower for their axes than for unit random vectors, ($p<0.01$ for all but one axes). In this regard, their method also discovers that the top eigenspace of PGGAN contains many informative transforms, and that alignment with dimensions of similar eigenvalues is better than mixing all the dimensions up.
>
> Note that, in their work, they chose to enforce orthonormality on the discovered axes for BigGAN noise space and StyleGAN FFHQ1024, which was not a constraint for the axes in PGGAN. In those two models, maybe due to orthogonality their axes do not have a uniform relationship with the Hessian structure of GAN. For example, in the BigGAN noise space, 3 out of 6 annotated axes have a significantly lower $vHv$ value (p<0.001) comparing to random vectors, namely these axes avoid the top eigenspace. Further, the same 3 out of 6 annotated axes have a significantly lower entropy in the top 15 dimension eigenspace ($p<0.05$) which suggests that although these axes avoid the top eigenspace, they are trained to concentrate power more on one of the top eigenvectors instead of mixing them randomly.
>
> This initial comparison has already provided some unexpected insights about the nature of interpretable axes. Further development is needed to draw a clear connection between the Hessian structure and the axes defined by in the previous works. This comparison encourages us to qualify our statements about unifying previous results, especially for Voynov Babenko 2020.

---

### Official Review · AnonReviewer3 · 2020-10-28
**Review: THE GEOMETRY OF DEEP GENERATIVE IMAGE MODELS AND ITS APPLICATIONS**

**Rating:** 6
**Confidence:** 3

**Review:**

This paper proposes a method for finding the axes of largest variation in the latent space of a generative model. This can be leveraged for better generator inversion and explainability. Several experiments are done to evaluate the latent vectors used by the method quantitatively and qualitatively.

##########################################################################

Reasons for score:

The analyses are interesting and the method seems novel. This method could prove useful in analysis of latent space properties. Its full significance and practicality could be clearer, though.

##########################################################################

Pros:
- The problem of understanding the latent space structure better is relevant and timely in the research on generative models equipped with a continuous latent space.
- Experiments seem to demonstrate successful dimensional truncation, GAN inversion, and that the eigenvectors behave consistently

##########################################################################

Cons:
- I am not sure I see whether each experiment actually supports the corresponding claim. Especially, the claim that interpretable axes are found seems central but it is not much supported in the experiments. It seems mostly represented in Fig. 9 in the Appendix, but I do not fully understand how we are supposed interpret the result in this figure.
- Fig 2 is hard to parse and clearly too tightly packed. Something should be done about it.

##########################################################################

Questions during rebuttal period:
- It is easy to get lost in the math and in the various measurements. Could the authors summarize, in practise and plain English, how their method should be used to find the most intepretable axes of variation for a new generative model, and how to confirm and measure that we have indeed found them?

Typo: on page 7 around "(Fig 5.)"

##########################################################################

Update after rebuttal discussions:
- In the light of the considerable overlap with [Chiu et al., 2020] pointed out by the other reviewers, I decreased my score. I have familiarized myself with it and can confirm the said overlap. However, given remaining differences, I do not find it unreasonable to consider this paper as "complementary" to [Chiu et al., 2020], *provided that the authors explicitly address the similarities in the final version*.
- I consider the sum total of contributions of the paper still tilting towards being sufficient for publication.

---

> ### Author Response · Authors · 2020-11-24
> **Response to AnonReviewer3: Additional MTurk Experiments Supports Interpretability; Tutorial on applying to new generators (1/2)**
>
> We appreciate the helpful and constructive reviews that the reviewers have made! We respond to each of the points below.
>
> > *I am not sure I see whether each experiment actually supports the corresponding claim. Especially, the claim that interpretable axes are found seems central but it is not much supported in the experiments. It seems mostly represented in Fig. 9 in the Appendix, but I do not fully understand how we are supposed interpret the result in this figure.*
>
> We thank the reviewer for this insight. We agree that in our initial draft, we were overreaching in our description of interpretability. In the two weeks since receiving this comment, we set out to formally investigate whether this method reveals axes that are perceptually relevant to individuals other than the authors. Using Amazon’s Mechanical Turk, we generated images under the identified Hessian directions of four different GANs (Progressive Growing GAN, BigGAN noise space, StyleGAN-Cat and StyleGAN-Face), and presented them to 175 participants, to investigate if the images were interpretable. We operationalized the concept of “interpretable” as follows: in each trial, we randomly sampled five reference images generated by a given GAN, and perturbed them (linearly or spherically) along the Hessian axis to be tested, which created five image sequences. These five sequences were shown on the same screen. The subjects viewed all the five sequences, and were asked if they could perceive a change, and if so, to describe the transformation that was common to the majority of the sequences. They were also asked to indicate how many sequences shared the identified transformation. Finally, subjects reported how large the perceived image change was (0%-100%), and the similarity of the image transformations across the five sequences (score of 1-9. 9 most similar) and the difficulty in interpreting the common change (InterpretDifficulty score on the scale of 1-9, 9 most difficult). Image perturbations comprised the top 10 eigenvectors, 10 random vectors orthogonal to these eigenvectors, and the bottom 10 eigenvectors in four GANs. Further, we added in photos of real objects as reference stimuli, which were manipulated by well-defined transformations such as rotation, perspective change, eye color, and more. Overall, subjects reported that the image sequences generated by top eigenvectors showed a larger amplitude of image change than both a) orthogonal directions (*t* = 3.4, *P* = 7.0 x 10^-4) and b) bottom eigenvectors (*t* = 6.4, *P* = 2.1 x 10^-10); notably, this was true even though we picked a much smaller step size in the top eigenspace than the orthogonal and bottom eigenspace. On average, the top 10 eigenvectors had a higher perceptual consistency score than a) the orthogonal-space random vectors (*t =* 2.8*, P =* 5.8 x 10-3) and b) the bottom eigenvectors (*t =* 4.4*, P =* 1.3 x 10-5). The subjects report that the top eigenvectors are easier to interpret than the bottom eigenvectors when they observe image change in the bottom eigenvectors. (*t =* -4.6*, P = 3.8* x 10-6). Thus, we have found stronger evidence for our initial report that these Hessian eigenvectors “capture perceptually relevant changes.” These results are now featured in Section 3.
>
> > *Fig 2 is hard to parse and clearly too tightly packed. Something should be done about it.*
>
> To improve this image, first, we lowered the density by enlarging the canvas of the figure; we also removed 1-2 GAN curves from each subplot, to improve legend readability.

---

> ### Author Response · Authors · 2020-11-24
> **Response to AnonReviewer3: Additional MTurk Experiments Supports Interpretability; Tutorial on applying to new generators (2/2)**
>
> > *It is easy to get lost in the math and in the various measurements. Could the authors summarize, in practise and plain English, how their method should be used to find the most intepretable axes of variation for a new generative model, and how to confirm and measure that we have indeed found them?*  *Typo: on page 7 around "(Fig 5.)"*
>
> Absolutely. To address this concern, first, we have re-written the Numerical Method section to provide a more plain-English description in the main text. Second, we will add a tutorial example after publishing our codebase. Basically, the steps are the following. When you get a new Generative model,
>
> 1. First wrap the Generator `G` with a wrapper class equipped with a `visualize` method. This method maps an array of hidden vectors to an array of images, and supports gradient flow between them.
> 2. Choose your favorite differentiable image dissimilarity metric `D`, which maps two images to a scalar and can compute gradient to it. We provide an interface to LPIPS, SSIM and MSE.
> 3. Pick a reference hidden vector `z` around which you would like to measure the metric tensor. Finally, send the object `G` `D` `z` into our `compute_hessian` function, then it will compute the eigenvectors, eigenvalues, and the full Hessian matrix, by various numerical methods as defined in the paper.
>    1. Depending on your need, if you want a few top axes to be computed quickly, you can choose `ForwardIter` and `BackwardIter` methods, which take a few seconds.
>    2. If you don’t care about time, and want a Hessian with full rank, you can choose the full `BP` method as described in the numerical methods, which takes one minute or a few minutes.
>    3. All 3 methods share the same interface, so you don’t need to worry about the numerical details.
> 4. After that you can visualize the change encoded by each of eigenvectors and see which of the vectors encode interesting transformations when applied to different reference images. In this final screening, we find our the axes that are likely to be interpretable.

---

### Official Review · AnonReviewer2 · 2020-10-28
**Significant overlap of the contributions with [Chiu et al., SIGGRAPH 2020]**

**Rating:** 5
**Confidence:** 4

**Review:**


Summary:

The paper proposes an analysis way of a latent space of GAN in a GAN architecture agnostic way. They use the Riemannian manifold analysis to investigate image manifold, which leads to a simple algorithm with eigen-decomposition of the Hessian matrix of a local point.

Unfortunately, many ideas and some of the findings (Hessian-based GAN exploration and anisotropy of Hessian in the latent space) are already well explored in [C1] with much higher quality and various modality applications.
But still, there are interesting ideas contained in this paper: 1) the latent space is homogeneous (in a sense that the major directions obtained by a latent position are shared at different positions with similar semantic meanings), and 2) extensive eigenvalue analyses of Hessian.

Based on the findings, they show interesting applications of efficient GAN Inversion, gradient-free search in Image space, and unsupervised discovery of interpretable axes. However, these applications are also already explored by [C1] in a visually pleasing way.

[C1] Human-in-the-Loop Differential Subspace Search in High-Dimensional
Latent Space, SIGGRAPH 2020.

Reasons for score:

I like the idea that leverages the findings of the homogeneous property of Hessian and improves the efficiency of GAN inversion with Hessian preconditioning. Also, I appreciate the analysis of eigenvalues and correlations.
However, unfortunately, most of the contributions the authors argued are largely overlapped with [C1], of which citation is missed by the authors. Thus, with these remaining contributions, this reviewer could not vote for the acceptance of this work at this point.
This reviewer would like to ask the significant different contribution from [C1] that this reviewer may miss.

Pros:
- The paper shows interesting interpretation and ideas, and its applications.
- The paper is written very clearly and readily.
- The authors validate the ideas with enough effort in the experiments.

Cons:
- The GAN architecture agnostic latent space exploration with Riemannian interpretation is already done in [C1] in a more general way.


Detail comments:
- While [C1] presents mainly about the SVD of Jacobian (but they also show Hessian interpretation (metric tensor) in Sec. 4.1 as well), it is essentially the same because of Eq. (3) in this paper. Also, [C1] suggested much more diverse exploration methods with diverse search methods.
Thus, this reviewer would like to listen to the authors' responses about this overlap.

---

> ### Author Response · Authors · 2020-11-19
> **Respond to AnonReviewer2. Our major contributions different from [Chiu et al., SIGGRAPH 2020] [1/2]**
>
> We thank our reviewer for the kind and constructive feedback. We were delighted to read this very recent Chiu et al. SIGGRAPH paper and were glad to find some overlap with our work, which we developed independently and in parallel. This overlap suggests an independent confirmation of the validity of the approach, which is exciting, and we have amended our manuscript to credit their important work. As pointed out by our reviewer, the SIGGRAPH numerical method bears some similarity to ours; the authors also applied their method to a problem reminiscent of those in our study (i.e. optimizing the black-box function of human perceptual judgment), and their Figures 3 and 4 also showed different rates of image change, proving the robustness of the geometric structure. However, there are strong differences between our studies: first, we provide an explicit, analytical link between the Jacobian of the generator to the Riemannian metric; second, we provide a numerical method that is faster and more accurate in approximating the full metric tensor, suitable to analyze the geometry of the latent space; third, we provide insights to the geometry through the homogeneity of the metric tensor. We elaborate on these points below.
>
> 1.  The SIGGRAPH paper’s mathematical framework for finding the most informative axes bears some similarity to ours. However, this article had no equation explicitly describing the connection between the Riemannian metric tensor and the Jacobian. In our reviewer’s example of section 4.1, Chiu et al. implied a Hessian interpretation of their approach, where the authors created a synthetic function using a quadratic form with an anisotropic spectrum to test their optimization method; but they did not elaborate on this interpretation for general generative models.  Our study presents this analytical connection explicitly — without our work, readers might not recognize this relationship and understand the geometric structure fully. We further went on to comprehensively measure the geometric properties of multiple state-of-the-art generative models, namely, we measured the local geometry of GAN latent spaces and their globalization.
>
> 2. In terms of the numerical method and efficiency, our approach differs substantially from that of Chiu et. al. Their numerical method computes the Jacobian matrix of the generator instead of the Hessian matrices of the squared distance. As recognized by the reviewer, if we have used L2 distance as the image distance function, then the inner product matrix of the Jacobian $H=J^TJ$ results in the Hessian. In the Jacobian matrix ($m$-by-$n$, sample dimension $m$, hidden dimension $n$), $m$ is usually much larger than $n$ (e.g. for a standard image generative network $n$ is ~200-500, and $m$ ranges from ~20,000-3,000,000), thus it is highly inefficient to compute a full Jacobian matrix as its complexity is O(m*single backprop time); in contrast, the complexity of our method is O(n * double backprop time) using common backpropagation. For example, for Progressive Growing GAN at 256 pixel-level, our method could obtain the full Hessian and its spectrum in 58 seconds, while the alternative method needs more than 4800 sec to obtain the spectrum of the full Jacobian. Thus, in their application, the authors approximate the Jacobian by randomly sampling output coordinates in sample space, and perform singular value decomposition (SVD) in the restrained space. This is equivalent to sampling rows of the Jacobian matrix to estimate its spectrum. In our approximation method, we perform the Hessian decomposition while doing backprop using Lanczos iteration, thus finding the largest eigenvalues and eigenvectors (equivalent to their largest right singular vectors).
>
> We conducted a numerical experiment to prove this. We showed that our numerical method approximates the real Hessian matrix faster and more accurately. Taking the Progressive Growing GAN with latent dimension n=512 as an example, using Lanczos iteration with finite-differencing method (ForwardIter), we can find the top 50 eigenvectors of the Hessian, approximating the full matrix to an accuracy of 0.99999 (per element-wise Pearson correlation) in 7.9 secs, while the alternative method, sampling 500 output dimensions, can only achieve 0.92420 correlation to the real Hessian in 12.5 secs. Thus, in terms of algorithmic efficiency, our method can compute the Riemannian metric tensor and the directions that are most informative faster, because the Lanczos iteration is more sample-efficient than a random sampling method (for more specific information, we will add this comparison to our Appendix). Although the alternative is a praiseworthy way to identify promising axes to explore the latent space of GANs, as the null space will not affect the exploration process in their application, our study provides a more efficient tool to interrogate the geometry of the GAN space.

---

> ### Author Response · Authors · 2020-11-19
> **Respond to AnonReviewer2. Our major differences from [Chiu et al., SIGGRAPH 2020] (2/2)**
>
> 3. In terms of the intended conceptual advance, our work focuses on analyzing the local and global geometry of the GANs, while the SIGGRAPH approach focuses more on improving the efficiency of optimizing a function over an image manifold created by the GAN. Although their paper presents a geometric interpretation, as noted by the Reviewer, we also analyzed extensively the homogeneity of the metric function over the GAN space. This homogeneity and anisotropy together provide a complete picture of the geometry of the manifold. Crucially, we developed a potential explanation about why this anisotropy might not have been reported in most other treatments of GAN latent space. This understanding of manifold geometry affords a multitude of insights about GANs, beyond facilitating optimization on GAN space with or without gradient. As one such insight, we predicted and showed that there should be some bad samples in the latent space when the latent vectors align with the top eigenspace, because as the images change at a much faster rate in the top eigenspace, even though these vectors live at a similar distance from the other vectors, the images they generate will be much farther away from the image distribution, so much so that the samples will look unnatural. If the metric tensors at different points are not well aligned (not homogeneous), then this prediction may not be true. (for details see the added section in revision). Also, we examined the non-linear PCA hypothesis of GANs and analyzed the correspondence of the sample space principal components (PCs) and the input space directions. And indeed, these sample-space PCs are more concentrated within the top eigenspace (although it's not a one-to-one relationship).
>
> Finally, per the issue of our scope being more limited relative to the SIGGRAPH paper, our updated manuscript shows our method is comparable in scope to the SIGGRAPH approach insofar as we are not limited to the image domain either. We applied our approach to WaveGAN (a sound generative networks based on DCGAN), using a spectrogram distance function for generated samples, and analyzed the geometry of its hidden space. We found that it also exhibits the homogeneity and anisotropy characteristic of the image GANs. Moreover, the optimization method in the SIGGRAPH paper is based on sequential line search with top eigenspaces visited more frequently; our optimization uses gradient-based (ADAM) or gradient-free (CMAES) methods assisted by the Riemannian metric. Thus, we believe that our approaches are complementary, showing that geometric information could be used in multiple ways to help optimization, either in gradient-based search, sequential-line search or CMAES-based methods.

---

> > ### Comment · AnonReviewer2 · 2020-11-23
> > **Thanks for the response, but the point could be misled**
> >
> > Thanks for elaborating the detailed difference from [Chiu et al.].
> > Overall, while there are some differences in detail algorithm parts, this submission indeed has considerable overlap in technical contribution and the main task.
> > From the authors' response, in my understanding, the main difference is on the emphasis of latent space geometry properties, i.e., anisotropy and homogeneity. (Other differences the authors defended are either deducible points or not a critical point that may flip the decision)
> > Thus, the key issue is whether the message this work conveys is substantial enough over the previous work.
> >
> > This reviewer agreed that anisotropy and homogeneity are interesting, but it has been known in the community (E.g., Homogeneity has been discussed as an analogy relationship in latent spaces, which is spontaneously obtained in distributed representation, e.g., [Mikolov 2013]. Anisotropy can be deducted in any work, including both this work and [Chiu et al.] once a low-rank approximation decision is made). Thus, the contribution of this work should be viewed as organizing previously known properties for recently emerging GANs in an interesting view.
> >
> > Therefore, we, reviewers, are suggested to assess whether these reframings are significant enough over the competing work [Chiu et al.].
> >
> >
> > <Detail comment>
> > - It is a false statement that Lanczos iteration is more sample-efficient than a random sampling method (see [Halko 2012]).
> > - Furthermore, the efficiency and approximation quality of eigenvectors should not be a concern of this work in that it is not a contribution of this work (typically, eigenvector directions are well approximated by many other randomized methods).
> > - Also, principal components and largest eigenvectors of Hessian and Jacobian boil down to the same thing in the context.
> >
> >
> > [Mikolov 2013] Mikolov, Distributed Representations of Words and Phrases and their Compositionality, NIPS 2013.
> >
> > [Halko 2012] Halko, Randomized methods for computing low-rank approximations of matrices, 2012.

---

> > > ### Author Response · Authors · 2020-11-25
> > > **Response to Anon Reviewer 2  (1/2)**
> > >
> > > > *Thanks for elaborating the detailed difference from [Chiu et al.]. Overall, while there are some differences in detail algorithm parts, this submission indeed has considerable overlap in technical contribution and the main task. From the authors' response, in my understanding, the main difference is on the emphasis of latent space geometry properties, i.e., anisotropy and homogeneity. (Other differences the authors defended are either deducible points or not a critical point that may flip the decision) Thus, the key issue is whether the message this work conveys is substantial enough over the previous work.
> > > > **This reviewer agreed that anisotropy and homogeneity are interesting, but it has been known in the community (E.g., Homogeneity has been discussed as an analogy relationship in latent spaces, which is spontaneously obtained in distributed representation, e.g., [Mikolov 2013]. Anisotropy can be deducted in any work, including both this work and [Chiu et al.] once a low-rank approximation decision is made). Thus, the contribution of this work should be viewed as organizing previously known properties for recently emerging GANs in an interesting view.** Therefore, we, reviewers, are suggested to assess whether these reframings are significant enough over the competing work [Chiu et al.].*
> > >
> > > The reviewer’s suggestion to reframe our manuscript as an organizing framework is well-taken. Indeed it is true that some of the key ideas presented in our manuscript have been described in different contexts - for example, we acknowledge that homogeneity is suggested by analogy in natural language processing models, visual analogy mentioned in the DCGAN paper, and anisotropy as implied in some figures in previous work on GANs. It is also true that some individuals could deduce or infer many points we have made explicit in our work. We believe that the value of our contribution is A) to provide an easy-to-use tool which is designed for and could be rapidly applied to analyze the geometry of generative models (we have applied it to audio generative models), B) to show explicitly how interesting research threads can come together to understand generative models and advance their applications. In this framing, we do not consider the Chiu *et al.* paper to be a competing work, but a complementary one. Further, while we do not doubt that many individuals (such as our reviewers) could deduce some of our results from previous literature, the readership that cares about GANs has grown increasingly diverse, encompassing scientists from computer science, biology and psychology; such a wide readership requires manuscripts that successfully and explicitly synthesize concepts in multiple threads of research.
> > >
> > > In addition to this synthesis, we believe our paper brings value to the field in that when we fully appreciate the anisotropy and homogeneity of that space, some points become obvious in the light of geometry. For example, we discover a large subspace GAN filled with small eigenvalues using the Hessian decomposition approach, while other researchers have figures conveying this in their work. However, when the homogeneity of the space is appreciated, this local tangent space with close to 0 eigenvalues becomes a global subspace, which is an effectively perceptual null space in the GAN (this is supported by our added Mechanical Turk experiments) The existence of this subspace suggests that GAN inversion will yield non-unique answers -- as the images are dominantly determined by their latent codes’ composition in space orthogonal to the null space. A related implication is that the interpretable axes defined in the GAN will not be unique. In previous work, the angles between latent axes have been used to compare the similarity of the interpretable axes (in \ref{Yujun Shen2020ClosedForm} Table 1), however, when the two axes are differed by a vector in the null space, they encode the same transforms semantically, but with a different speed of change. Thus we suggest that making the underlying geometry clear is very helpful to works finding interpretable structures in GANs.
> > >
> > > Furthermore, this framework is applicable to multiple applications regarding optimization or exploration in the GAN space. Readers of the Chiu et al. paper may not realize that the forest exists when they are focused on the application of human-in-the-loop interaction. In the revision, we performed more numerical experiments to show that the eigenbasis preconditioning could be applied to multiple GANs and datasets (StyleGAN Face, PGGAN CelebA, inverting FFHQ and CelebA) and consistently improve their approximation of real images.
> > >
> > > Shen, Y., & Zhou, B. (2020). Closed-form factorization of latent semantics in gans. *arXiv preprint arXiv:2007.06600*.

---

> > > ### Author Response · Authors · 2020-11-25
> > > **Re- Respond to AnonReviewer2 (2/2)**
> > >
> > > >*Detail comment - It is a false statement that Lanczos iteration is more sample-efficient than a random sampling method (see [Halko 2012]). - Furthermore, the efficiency and approximation quality of eigenvectors should not be a concern of this work in that it is not a contribution of this work (typically, eigenvector directions are well approximated by many other randomized methods). - Also, principal components and largest eigenvectors of Hessian and Jacobian boil down to the same thing in the context.*
> > > >*[Mikolov 2013] Mikolov, Distributed Representations of Words and Phrases and their Compositionality, NIPS 2013.*
> > > >*[Halko 2012] Halko, Randomized methods for computing low-rank approximations of matrices, 2012.*
> > >
> > > This is a good point. In that sentence, we were referring to the specific numerical experiments we did above, rather than the general characteristic of these two classes of algorithms. For the purpose of that practical demonstration, we were probably not using the best random sampling methods, although we do not yet have a practical way to identify these best methods. As suggested by the thesis, in a general matrix approximation problem, many optimizations could be made for random sampling methods to increase efficiency, which may or may not be applicable to the case of Hessian of Jacobian matrix approximation for a neural network. For example, we did not take parallelization (computing multiple columns of Jacobian in one forward-backward pass) into consideration in that demonstration, which is a general way to accelerate random sampling approaches that may make their methods’ speed comparable to ours. We will be happy to compare the random sampling method implemented in the SIGGRAPH paper once that code is shared.
> > >
> > > Finally, the relevant thesis work was a great reference for this topic, we are appreciative of the note.

---

### Author Response · Authors · 2021-03-18
**Thanks for the decision! Added comparison with Peebles et al.**

The authors really appreciate the reviewers' and AC's effort for the nice comment and the final decision.

As requested, in the camera-ready version, we added a supplementary section making a detailed comparison of our work and Peebles et al. . In short we showed that the interpretable axes found by Peebles et. al. have a striking correspondence with our top Hessian eigenspace, meanwhile our approach is much more efficient in discovering these axes. However, the stochastic estimator of Hessian diagonality as proposed in Peebles et al. is very useful as a regularizer in GAN training, while our more precise approach is more suitable for post hoc analysis of the geometry of GANs. We encourage readers to check out the new version.

---

### Decision · Program_Chairs · 2021-01-07
**Final Decision**

**Decision:**

Accept (Poster)

**Comment:**

This paper presents several analyses on the geometry of GAN generators through the lens of Riemannian geometry: showing interpretability of the leading eigenvectors of the Hessian, homogeneity of the space, and more efficient latent-space inference through preconditioning. Reviewers found the (revised) paper well-written and clear, with a thorough set of experiments to support their main claims. While there were several concerns around the generality of the approach, the authors performed several experiments in the rebuttal period to address many of the reviewer’s concerns (robustness of findings with different image distance functions, inversion on additional GANs and datasets, user study of perceptual properties of axes, and comparison to previous methods for intepretable axes discovery). I found these experiments extensive and convincing, supporting the claims around robustness of the approach to different image distance metrics, GAN architectures, and interpretability of the axes.

There were also strong concerns around similarity with recent work (Chiu et al., SIGGRAPH 2020 and Peebles et al., ECCV 2020), but both of these papers were published at most 1 month before the ICLR submission deadline, and thus should be considered as concurrent work.

Given the strong set of additional experiments and interesting empirical observations, I recommend accepting this paper.

There remain concerts around the extent to which the findings “unify” previous approaches on interpretable axes, and we encourage the authors to update the paper before the camera ready to address these and additional reviewer concerns (especially expanding the discussion of the relationship with concurrent work in Chiu et al. and Peebles et al.).